

# Risk-based flood protection planning under climate change and modelling uncertainty: a pre-alpine case study

Beatrice Dittes[1], Maria Kaiser[2], Olga Špačková[1], Wolfgang Rieger[2], Markus Disse[2], Daniel Straub[1]

[1]Engineering Risk Analysis Group, Technische Universität München, München, 80333, Germany
[2]Chair of Hydrology and River Basin Management, Technische Universität München, München, 80333, Germany

*Correspondence to*: Beatrice Dittes (beatrice.dittes@tum.de)

## Abstract

Planning authorities are faced with a range of questions when planning flood protection measures: is the existing protection adequate for current and future demands or should it be extended? How will flood patterns change in the future? How should
the uncertainty pertaining to this influence the planning decision, e.g. for delaying planning or including a safety margin? Is it sufficient to follow a protection criterion (e.g. to protect from the 100-year flood) or should the planning be conducted in a risk-based way? How important is it for flood protection planning to accurately estimate flood frequency (changes), costs and damages? These are questions that we address for a medium-sized pre-alpine catchment in southern Germany, using a sequential Bayesian decision making framework that quantitatively addresses the full spectrum of uncertainty. We evaluate
different flood protection systems considered by local agencies in a test study catchment. Despite large uncertainties in damage, cost and climate, the recommendation is robustly for the most conservative approach. This demonstrates the feasibility of making robust decisions under large uncertainty. Furthermore, by comparison to a previous study, it highlights the benefits of risk-based planning over a planning of flood protection to a prescribed return period.

## 1   Introduction

Technical flood protection measures have long life times of, on average, 80 years (Bund / Länder-Arbeitsgemeinschaft Wasser, 2005). The uncertainty over such a long planning horizon is large, both in terms of climatic and socio-economic development. It is thus not trivial for planning authorities to take decisions on flood protection planning that are economical while not leading to excessive losses or high adjustment costs. I, it is important to consider costs – in construction, adjustment and flood damages – over the entire measure life time.

Ideally, the planning of flood protection infrastructure is performed through a risk-based approach. Thereby, potential damages are considered in the decision-making process. Considering that the annual maximum discharge $Q$ is the main driver for flood damages, the annual flood risk in year $t$, $r_t$, is defined as (e.g. Merz et al., 2010a)



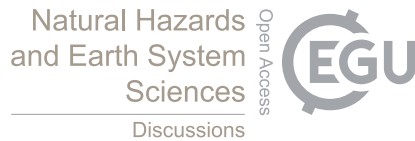

$$r_{\mathrm{t}} = \int\limits_{0}^{\infty} f_{\mathrm{Q}}(q) d_{\mathrm{t}}(q) \mathrm{d}q \tag{1}$$

where $f_{\mathrm{Q}}(q)$ is the probability density function (PDF) of the annual maximum discharge and $d_{\mathrm{t}}(q)$ is the damage associated with the flood discharge $q$ in year $t$. If decisions are based on a Cost-Benefit Analysis (CBA), the optimal flood protection strategy $s$ is the one that minimizes the sum of risks and costs over the life-time of the protection system (Špačková and Straub, 2015):

$$s^{\mathrm{opt}} = \arg\min_{s}\big(c^{\mathrm{tot}}(s) + r^{\mathrm{tot}}(s)\big), \tag{2}$$

where $c^{\mathrm{tot}}(s)$ and $r^{\mathrm{tot}}(s)$ are the expected costs and risks associated with strategy $s$. They are net present values, discounted to the time of planning (viz. Eq. (5)).

In contrast, the current practice in most European countries is to require protection from floods of a certain return period, often the 100-year flood (Central European Flood Risk Assessment and Management in CENTROPE, 2013). This simplifies planning, as it is not necessary to quantify damages. While a well-chosen criterion may lead to near-optimal

strategies, a suboptimal one may lead to strategies that are too risk-averse or too risky. To balance the optimality of risk-based planning with the lower effort of criterion-based planning, authorities can apply a 'zoning' approach, with different regions being assigned a different protection criterion. However, (Kind, 2014) performed CBA in Dutch catchments and found that despite the applied large-scale zoning, the fixed protection criterions were not economically efficient, because of both over- and underprotection.

There is a growing consensus that costs and damages (and thus the extreme discharge causing them) should be modelled probabilistically (Aghakouchak et al., 2013). Probabilistic modelling allows to include the (large) uncertainty in future climate change in the CBA. The climatic uncertainty is included in the PDF of the annual maximum discharge $f_{\mathrm{Q}}(q)$ (viz. Eq. (1)), which is expected to change depending on future climatic developments. Recent studies have aimed at quantifying the climatic uncertainty components in (extreme) discharge and precipitation, (Bosshard et al., 2013; Hawkins and

Sutton, 2011; Sunyer, 2014).

Risk-based flood protection planning has a long history (Clark, 1980; James and Hall, 1986; Lund, 2002) and approaches to account for uncertainty in the risk estimate have also been developed for some time (Kundzewicz et al., 2010; USACE, 1996). Recent fields of interest in risk-based flood protection planning are e.g. how to include the flexibility of measures into the decision making (i.e. how costly it is to adjust measures later on) (Klijn et al., 2015; Kuklicke and Demeritt,

2016; Woodward et al., 2014) and how to account for non-stationarity in discharges and risk estimates, e.g. due to climate change (Rehan and Hall, 2016; Rosner et al., 2014; Sayers et al., 2013). We have recently presented a framework for flood protection planning that is based on a quantitative, probabilistic joint estimate of climatic (and potentially other) uncertainties, naturally incorporating non-stationarity and the flexibility of the protection system in a Bayesian framework (Dittes et al.,





2017a, 2017b). In this work, we apply the framework to a real decision making problem of four potential protection strategies as currently considered by the local water management authorities. To our knowledge, this is the first risk-based application of a fully quantitative, continuous (not scenario-based) decision making framework for sequential flood protection planning that probabilistically includes future decisions and discharges. We additionally compare the results from this risk-based

5    planning to previous results from a criterion-based planning for the same site. We find significant differences, which lead us to discuss the virtues of the two alternative planning paradigms.

We provide an overview of the applied uncertainty quantification and decision making methodology in Sect. 2. In Sect. 3, we give details of the case study, outlining the considered strategies, the modeling of flood events and the damage assessment. Results are presented and discussed in Sect. 4, followed by the conclusions in Sect. 5.

## 10    2    Theoretical background

In this section, we provide a short overview of the proposed methodology for flood protection planning under uncertainty. In Sect. 2.1, we outline how the climate-related uncertainty in discharge ($f_Q(q)$ in Eq. (1)) is defined and quantified for use in decision making, following (Dittes et al., 2017a). In Sect. 2.2, we summarize the Bayesian sequential decision framework for optimizing the flood protection strategy under uncertainty.

## 15    2.1    Uncertainty in determining future extreme discharges

In practical planning applications, there are only limited data and models available for estimating future flood magnitude and frequency, which leads to uncertainty in predictions. The problem is exacerbated when climate change effects (i.e. non-stationarity, or trends) are present in this limited information, since it means more parameters need to be learned from a given, limited set of data. When planning under uncertainty, it is important not to consider different uncertainties (from climate, lack

20    of data, etc.) individually, but jointly (Sunyer, 2014).

This section provides a short overview on how the uncertainty in future extreme discharges can be modeled from limited data, taking into account non-stationarity and considering uncertainties jointly. The data consists of a historic record of discharges and an (imperfect) ensemble of discharge projections that are a result of climate modelling. The former are often of limited length, whereas the latter may exhibit bias and significant uncertainty that builds up over the climate modelling

25    chain (starting from global emission pathways down to downscaled values for a particular study catchment).

The PDF of annual maximum discharges, $f_Q(q)$, is modeled mathematically as an extreme value distribution, e.g. the Gumbel or the Generalized Extreme Value (GEV) distribution. These distributions are parametrized by a set of parameters $\boldsymbol{\theta}$. Non-stationarity in the discharge estimate is modelled by including additional parameters in $\boldsymbol{\theta}$ describing the time-dependence. The distribution of the parameters $\boldsymbol{\theta}$ can be learned by Bayesian analysis from Y years of annual maximum discharges





$q = [q_1, \dots, q_Y]$. Assuming independence between annual maxima of different years, the posterior PDF of the parameters $f_{\Theta|Q}(\theta|q)$ is obtained as

$$f_{\Theta|Q}(\theta|q) \propto f_{Q|\Theta}(q|\theta) \times f_{\Theta}(\theta) = \prod_{t=1}^{Y} f_{Q|\Theta}(q_t|\theta) \times f_{\Theta}(\theta), \qquad (3)$$

where $f_{\Theta}$ is the prior PDF of the parameters, $f_{Q|\Theta}$ is the PDF of the applied extreme value distribution wherein the dependence on $\theta$ has been made explicit, and $f_{Q|\Theta}(q|\theta) = \prod_{t=1}^{Y} f_{Q|\Theta}(q_t|\theta)$ is the likelihood describing the data. In the equality, we assume

independence among the annual discharge maxima $q$.

Learning the PDF of $\theta$ from projections is more intricate since uncertainties from climate modelling must be accounted for. For use in flood protection planning, we categorize the climatic uncertainties as follows (Dittes et al., 2017a):

- ‘Visible uncertainty’, which is known and can be quantified. For an ensemble of discharge projections, this would e.g. be the internal variability (natural fluctuations) and the model response uncertainty (also known as the spread of
the ensemble).

- ‘Hidden uncertainty’, which is the remaining uncertainty and can, at best, be estimated. E.g., in the projection ensemble of the case study, the uncertainty in the emission forcing is hidden since all projections are based on the same emission scenario. In real planning situations, hidden uncertainty is typically significant because of limited and imperfect projections and data, it can therefore not be neglected.

In Fig. 1, we show the hidden uncertainty and internal variability over the projection horizon for one particular projection (CCLM, viz. Sect. 3.3).

Including the hidden uncertainty via its standard deviation $\sigma_t^{\text{(hidden)}}$, the likelihood $f_{Q|\Theta}(q|\theta)$ of Eq. (3) becomes:

$$f_{Q|\Theta}(q|\theta) = \int_{-\infty}^{\infty} \left[ \prod_{t=1}^{Y} f_{Q|\Theta}\big(q_t - z \times \sigma_t^{\text{(hidden)}}|\theta\big) \right] \times \varphi(z)\, \mathrm{d}z, \qquad (4)$$

where $z$ is a random variable following the standard normal PDF $\varphi$. Visible uncertainties are included in different ways, e.g. the internal variability is a natural component of Eq. (4) through $q_t$, whereas the ensemble spread is inherent in combining the
parameter PDFs $f_{\Theta|Q}(\theta|q)$ from different members of a projection ensemble. For this combination, we apply the concept of effective projections (Pennell and Reichler, 2011; Sunyer et al., 2013), whereby a projection ensemble is split into multiple sets of ‘effective projections’. We multiply the PDFs of the members within one set and average in between sets to obtain a joint parameter PDF. The rationale and details of the implementation can be found in (Dittes et al., 2017a).

We take the pragmatic approach of using only discharge projections for initially estimating the joint parameter PDF
$f_{\Theta|Q}(\theta|q)$, not the available historic record. This is motivated by the fact that the historic discharge has been used to bias-





correct the projections; using historic record alongside projections would thus correspond to a double counting. In practice, however, there will often be a discrepancy between the projections and historic discharge even for the reference period, as the projections may have been corrected to match means and not extremes or to match a larger geographical region. Some mismatch between projections and historic record is present also in the catchment studied here, as will be discussed in Sect. 5.

**2.2   Framework for risk-based optimization of flood protection under uncertainty**

Flood protection is a dynamic process, as illustrated in Fig. 2: A flood protection system is implemented initially and later revised, based on data (e.g., discharge observations) that becomes available in the future. Future discharges are uncertain and the damages they cause will depend on the protection system in place. The expected damages are the risks. The sum of the two monetary quantities, risks and costs, is to be minimized over the measure life-time (viz. Eq. (2)). If the demand has changed

based on the new observations, it may be necessary or desirable to adjust the protection capacity. The cost for both the initial implementation of the protection system and for adjustments depends on the system flexibility: a more flexible system decreases adjustment costs, but this saving must be balanced with potentially higher costs of implementing a flexible system initially. When there is large uncertainty, it becomes more likely that a design has to be adjusted later on, as more information becomes available. To take these aspects into account, we have developed a quantitative decision framework that considers

planning as a sequential process. It accounts for the system flexibility and the future learning process through Bayesian updating of the initial PDF of parameters, $f_{\Theta|Q}(\theta|q)$ (viz. Sect. 2.1.), with new information in the future (Dittes et al., 2017b). It evaluates, which flood protection system is recommendable based on the uncertainty in extreme discharge, described by $f_{\Theta|Q}(\theta|q)$, and the flexibility of the considered flood protection systems. As will be shown in Sect. 3.5, the flexibility is intrinsic in the measure costs in this case study.

The updating with future information is done probabilistically and the protection recommendation is found via backwards induction optimization, taking into account the possible future adjustment decisions. Backwards induction optimization (Raiffa and Schlaifer, 1961) works by first determining the system that should be installed at the last adjustment, depending on the existing protection and observations (data) available by then. The obtained recommendation is then used to find the system that should be installed at the second to last adjustment and so forth until arriving at a recommendation for the

system that should be installed initially.

In (Dittes et al., 2017b), the optimization was presented for a criterion-based approach to flood protection planning (e.g. protecting from the 100-year flood). Here, we use a risk-based approach, i.e. we consider damages and optimize for the best balance of residual risks and costs (viz. Eq. (2)) instead of relying on a protection criterion. The total risk associated with a strategy is

$$r^{\text{tot}}(v_1, \ldots, v_{N \times \Delta t}) = \sum_{t=1}^{N \times \Delta t} \frac{r_t(v_t)}{(1 + \lambda)^t},$$  (5)





where $v_t$ is the capacity of the protection system at time $t$. The annual rate of discounting is $\lambda$ and the time period between decisions in years is $\Delta t$. The cost $c^{\mathrm{tot}}$ is defined analogously. Including the risk in the analysis in addition to the cost can easily be done in a setup with few possible strategies.

Note that the simplicity of Eq. (5) is deceiving. In particular, the analysis is not static. Instead, risks (and costs) are evaluated
probabilistically for all strategies and the best strategy is found inductively as described, by backwards induction starting at the final revision of the flood protection system. We illustrate the corresponding planning process in Fig. 3. Due to computational limitations, for a set-up with a larger number of decision steps, it may be necessary to use a POMDP approach instead – as has been described e.g. in ( Špačková and Straub, 2017).

## 3     Case study Rosenheim, Germany

In this study, we apply the concepts outlined in the previous section to the municipality of Rosenheim in Bavaria, Germany. Rosenheim closely surrounds the Mangfall river on both sides, making it vulnerable to flooding. An extreme flood event in 2013 underlined the need for an improved flood protection concept for Rosenheim.

### 3.1    Description of study area

The Mangfall catchment (Fig. 4) has a size of about 1,100 km² and is situated in southern Germany, in the Bavarian pre-Alps.
The Mangfall River is a medium-sized river, whose yearly average discharge at the gauge in Rosenheim is 18 m³ s⁻¹; the official estimate of the 100-year discharge at that gauge is 480 m³ s⁻¹ (Hochwassernachrichtendienst Bayern, 2017; Wasserwirtschaftsamt Rosenheim, 2017).

In the east of the Mangfall catchment lies the city of Rosenheim with more than 60,000 inhabitants, which experienced numerous flood events. The largest flood event so far occurred in 1899, for which a discharge of approximately 600 m³ s⁻¹
was reconstructed (Wasserwirtschaftsamt Rosenheim, 2017). The second largest flood occurred in 2013 with an extreme discharge of 480 m³ s⁻¹ in Rosenheim (Wasserwirtschaftsamt Rosenheim, 2014). Damage estimates for the 2013 flood range from €150 to €200 M for Rosenheim and the neighboring city of Kolbermoor (Wasserwirtschaftsamt Rosenheim, 2014, 2017).

As a consequence of the 2013 flood event, the flood protection along the Mangfall River is currently being improved. After completion of the new flood protection system in 2020, dikes and embankments in Rosenheim should withstand a design
discharge of 480 m³ s⁻¹ (plus freeboard). In addition, a flood polder 23 km east of Rosenheim is planned in order to compensate for the discharge aggravation and the loss of retention area due to the dike expansion along the Mangfall River (Wasserwirtschaftsamt Rosenheim, 2017). Within the municipal area of Rosenheim, dikes and walls protect the residents from flooding by the Mangfall River and two creeks (Fig. 4).

### 3.2    Investigated flood protection systems

We investigated four protection systems for the risk-based flood protection planning of the city of Rosenheim:



S1: protection system 2020 – current state (as it is implemented)

S2: protection system 2020 + flood polder

S3: elevation of protection system 2020 by 1 m

S4: elevation of protection system 2020 by 1 m + flood polder

The first flood protection system (S1) represents the protection system in Rosenheim that will be completed by the year 2020 (Fig. 5). It consists of structural protection measures along the Mangfall River and the two creeks Auerbach and Kaltenbach. Homogeneous dikes and dikes with inner cores closely surround the Mangfall River on both sides. Where the building development allows, the dikes are set back from the river to increase the retention area. While dikes and walls completely embank the Auerbach on both sides, the Kaltenbach only borders a dike on the right, thus allowing the flooding of the adjacent

forest on the left. In sum, 16 kilometers of dikes protect the urban area of Rosenheim. In addition, about 1 km of walls protect the residents around the Auerbach. The freeboard of the structural protection measures is 1 m with two exceptions. The first exception is the dike along the Kaltenbach, which has a freeboard of 0.5 m. The second exception is a dike section with 0.4 m freeboard near the mouth in the Inn (Fig. 5). Overall, S1 withstands a design discharge of 480 $m^3 s^{-1}$.

    The second flood protection system (S2) combines the protection system 2020 and a flood polder upstream on the

Mangfall River (Fig. 5). Planned as a controlled retention basin, the flood polder is used when the predicted runoff for the gauge Rosenheim exceeds the design discharge of S1. Then, the 12 floodgates with a width of 4 m and a maximum lift height of 1.5 m are opened (RMD Consult, 2016). The floodgate control depends on the shape and the volume of the predicted flood wave. In use, the flood polder can store a maximum volume of 6.62 million $m^3$ (RMD Consult, 2016).

    The third flood protection system (S3) further increases the safety level of S1 by elevating the existing protection

measures. The dikes and protection walls of S1 are heightened uniformly by 1 m, as this increase is considered to be still technically feasible. Due to limited space available, a widening of the dike base in order to elevate the embankments is not possible. Instead, a dike elevation can only be achieved by means of steeper dike slopes or protection walls on top. As the dikes in Rosenheim are already quite steep (with slopes between 2/5 and 1/2), a dike with a wall on its crest would be the preferred solution. The local water management office assumes that the current static of the protection walls would not

withstand the additional hydrostatic pressure of a 1 m elevation. Therefore, the protection walls would have to be rebuilt for the implementation of S3, making the extension of S1 to S3 a non-flexible strategy.

    The fourth flood protection system (S4) provides the greatest protection for Rosenheim of all systems, as it combines the elevated protection system 2020 and the flood polder.

## 3.3  Available discharge projections

At the gauge Rosenheim at the Mangfall, an ensemble of discharge projections is available up to the year 2098. Table 1 lists the projections of the ensemble. The projections are available as daily means. In this case study, we use only the annual maxima





of the daily means, as reproduced in the supplement. In Fig. 6, we show the probability of a flood event exceeding S1 (480 $m^3s^{-1}$), according to each individual projection. The exceedance probability is shown at four points in time: initial planning (year 0), as well as at years 30, 60 and 90. For reasons that will be described in Sec. 3.6, these points are defined to correspond to years 2008, 2038, 2068 and 2098, respectively. Results shown at the individual points in time are calculated by using

projections from 1970 (the year in which the historical record starts) up to the year in question. The figure highlights the spread of the ensemble as well as the fact that, at up to 4% annually, the chance of exceeding S1 is projected to be quite high, so there likely is a need for further protection.

Several of the projections have identical modeling chains and differ only in the model run, six of the ten Regional Climate Models are nested into the same Global Climate Model, ECHAM5, and all GCM-RCMs are based on the same SRES forcing

scenario (A1B), coupled to the same hydrological model (WaSiM v8.06.02 at a resolution of 1km²) and same downscaling technique (quantile mapping). The ensemble is limited in that it does not cover a wide range of modeling uncertainties, and it is imperfect in that the projections of the ensemble are not independent. To account for this dependence, following the results of (Sunyer et al., 2013), we partition the ten available projections into two sets of five 'effective projections' (viz. Sect. 2.1 or, for more details, (Dittes et al., 2017a)). To do so, we consider genealogy: projections with a very similar modeling chain should

be in different sets to provide the maximum amount of independent information per set. The result is:

- ▪ Set 1: CLM1, CCLM, REMO2, HadGM, RACMO;
- ▪ Set 2: CLM2, REMO1, REMO3, HadRM, BCM.

We used all available future discharge projections to learn the joint annual maximum discharge projection PDF. This is based on the premise that it is good to use a projection ensemble that is as large as possible, even if some projections may be

a less faithful representation of the truth than others (Huang et al., 2014; Knutti, 2010; Reifen and Toumi, 2009). We do not weigh projections since there is an ongoing debate about this. If desired, it would be straightforward to introduce weights into the analysis.

### 3.4   Hydrodynamic modeling of flood events

The estimated flood damages for the optimization framework resulted from a sequence of modeling and calculation steps. By

means of a hydrodynamic model, we simulated flood events with peak discharges ranging from 520 to 740 $m^3 s^{-1}$. We modeled inundation depths and flood extents for four different flood protection systems with the two-dimensional hydrodynamic model HYDRO_AS-2D (Nujic, 2003). The simulated inundation depths then served as inputs to a meso-scale flood damage model. For the damage estimation we used the Rhine Atlas model (RAM) developed by the International Commission for the Protection of the Rhine (ICPR) (IKSR, 2001) with two different land cover data inputs. For further comparison, we additionally

applied a simple damage assessment method (SDAM) by the Bavarian water management administration.

We modified an existing hydrodynamic model of the Mangfall River by (Nujic and Blasy-Øverland, 2005). In the model, cross-sectional profiles, supplemented by terrestrially measured riverbank lines, represent the geometry of the Mangfall



River and its tributaries. The roughness values of the floodplain were determined based on the land use type, which was derived from aerial images on a scale of 1:5,000. Observed water levels of a flood event in 2001 served the calibration of the Mangfall River's channel roughness. The 2D-model also represents constructions such as bridges and structural protection measures, which are relevant for the appropriate flood modeling (Nujic and Blasy-Øverland, 2005).

Since buildings oppose a resistance to the surface flow, they can strongly influence the water level and thus the flood forecasting and damage estimation (Schubert et al., 2008; Schubert and Sanders, 2012). The original 2D-model of the study site, however, disregards buildings. Therefore, we incorporated the buildings into the model based on OpenStreetMap data. In addition, we disabled the building elements in HYDRO_AS-2D to prevent an overflow of the building elements and thus reflect the building blockage effect in the simulations.

For the flood modeling, we further updated the 2D-model to match the basic protection system S1. As the hydrodynamic model does not cover the flood polder, we considered its effect indirectly in the simulation via the input hydrograph. Due to the controlled water extraction, the polder causes a capping of the flood wave. We reflected this effect in the model by reducing the input flood wave by the maximum retention volume of the polder. Hence, to simulate S2, we applied the mesh of S1 in combination with the cut input hydrographs. For the simulation of S3, we elevated the protection measures

in the model by 1 m assuming the same cross-sections as before. For the simulation of S4, we ran the hydrodynamic model of S3 with the cut input hydrographs of S2.

## 3.5   Flood damage assessment

Uncertainties are inherent in hydrodynamic modeling as well as in damage modeling. However, various authors believe that the uncertainty of the simulated water depths is low compared to the uncertainties resulting from the choice of the depth-

damage-functions and the specific asset values (Apel et al., 2009; de Moel and Aerts, 2011). The differences among damage estimates of different damage models for the same flood event are large. Comparative studies of flood loss models demonstrate this (Apel et al., 2009; Cammerer et al., 2013; Jongman et al., 2012; Wünsch et al., 2009). We hence use three different damage models for the optimization, introduced in Sect. 3.5.1 to 3.5.3, and validate their results against the flood event of 2013 in Sect. 3.5.4.

### 3.5.1   Rhine Atlas model (RAM)

The Rhine Atlas model (RAM) (IKSR, 2001) is a meso-scale flood loss model that uses relative stage-damage functions for aggregated land cover categories. The stage-damage curves are derived from empirical damage values from the German flood loss database HOWAS (Merz et al., 2004) and expert judgements (IKSR, 2001). The model assesses flood damage for the following five land use types: settlement, industry, infrastructure, agricultural land, and forestry. For the three built-up area

types, the RAM also distinguishes between mobile and immobile assets (IKSR, 2001). The RAM calculates the flood damage as the multiplication of the estimated flood loss ratio and the specific value of the affected asset. As input parameter, the model requires land use and inundation depths. The output of the model is a raster grid indicating a damage value per grid cell.



Numerous studies have already applied the RAM. On the one hand, the model has been used for scenario calculation (Bubeck et al., 2011; de Moel and Aerts, 2011; Wagenaar et al., 2016). On the other hand, the RAM was applied in comparison studies in which the modeled losses were validated against observed flood damages (Apel et al., 2009; Cammerer et al., 2013; Jongman et al., 2012; Kellermann et al., 2015; Thieken, 2008; Wünsch et al., 2009). These comparative studies also highlighted

the shortcomings of the RAM. Some studies showed that the model tends to underestimate the reported flood losses (Apel et al., 2009; Jongman et al., 2012). More generally, as (Cammerer et al., 2013) point out, it is not readily possible to transfer a flood loss model to another geographical region than it was developed for. Therefore, using a damage model derived from data of similar flood and building characteristics as the study site is advisable, such as the SDAM (viz. Sect. 3.5.3.).

### 3.5.2    Adaptation of the Rhine Atlas model to the study area

To adapt the RAM to the regional situation, we adjusted the specific asset values based on the gross domestic product (GDP). With a GDP per capita of €43,520 Rosenheim surpasses the Bavarian average of €39,691 (VGR der Länder, 2015). Therefore, the asset values for Bavaria stored in the RAM were increased by 9.6 % (Tab. 2).

In this study, we applied the RAM with two different land use data sets. One version applies the CORINE land cover (CLC) data with a 100m × 100m resolution and another version uses the digital basic landscape model (Basic DLM) from the

German Authorative Topographic-Cartographic Information System (ATKIS) with a 1m × 1 resolution. Originally, the RAM allocates the specific assets and depth-damage functions based on the CLC data. The ATKIS data set identifies 21 land use classes in Rosenheim, whereas the CLC data set differentiates 12 land use classes (Tab. 3). Figure 7 illustrates the differences between the two land cover data sets regarding the spatial distribution and resolution of the five damage categories. Especially in the settlement area, the ATKIS data set more accurately distinguishes patches of traffic, industry and areas of other use.

Furthermore, the ATKIS also classifies the linear structures of the Mangfall River and the creeks discharging into it, which the CLC disregards. However, since the RAM was originally developed for the coarser raster of the CLC, the results obtained with ATKIS are not necessarily more accurate.

### 3.5.3    Damage assessment according to the Bavarian water management administration (SDAM)

In addition to the RAM, we applied a simple damage assessment method (SDAM) by the Bavarian water management

administration. The Bavarian water management administration uses this method in order to prioritize the implementation of flood protection measures based on the determined damage potential. For simplicity, the method assumes that the replacement cost for each affected building amounts to €50,000 times a usage factor reflecting the building category (Tab. 4). Accordingly, the damage potential $DP$ is calculated as follows

$$DP = \sum f_\mathrm{u} \cdot N_\mathrm{u} \cdot €50,000 \, , \tag{6}$$





where $f_u$ is the usage factor of the building category u, and $N_u$ is the number of affected buildings of the building category u. The local water management office provided us with the usage classification of the more than 18,000 buildings. The majority of the buildings in Rosenheim are residential houses (88 %), whereas businesses account for 6 % of the buildings. The remaining 6 % of the buildings belong to the categories public facilities, infrastructure, and special cases.

### 3.5.4    Validation of the damage models on the flood event in 2013

In June 2013, a major flood hit the city of Rosenheim. Due to the severity of the event, the inundation area was documented by means of aerial photographs. Unfortunately, there was no documentation of the flood losses. The only indication is a loss estimate of €150 to €200 M for Rosenheim and its neighboring city of Kolbermoor, with Kolbermoor being more heavily affected by flooding (Wasserwirtschaftsamt Rosenheim, 2014, 2017). For reasons of model validation, we simulated the flood event of 2013 and compared the results to the inundation area estimated from aerial images. Although the simulation underestimates the mapped flood extent by approximately 12 %, the simulation shows good agreement with the documented inundation in all essential areas of flooding. On this basis, we rate the simulation of the flood event 2013 as good within the scope of validation possibilities.

In the next step, we compared the reported flood loss range to the damage estimated by the RAM and the SDAM. As the city of Kolbermoor was more heavily affected by the flood 2013, we assumed – in consultation with the local water management office – that the losses in Rosenheim made up 25 to 35 % of the total reported damage. With a total loss sum of €150 to €200 M, the flood loss in Rosenheim amounts to about €38 to €70 M. As Tab. 5 shows, the RAM returns very different damage estimates depending on which land cover data set is used. Applying the CLC data set results in a damage estimate of €97 M, whereas with ATKIS the damage estimate amounts to €15 M. Application of the CLC data set overestimates losses because CLC attributes the Mangfall River erroneously to the category of settlement and industry (Fig. 7). Application of the ATKis data leads to a strong underestimation of the damage, due to the fine resolution of the ATKIS data, which is not consistent with the original calibration of the RAM. Among the investigated methods, the simple damage estimation method leads to damage estimates closest to the reported damage range. For 1,050 affected houses, the method estimates a damage of €78 M. The optimization was conducted for RAM with both land cover sets as well as SDAM.

### 3.6    Optimization setup and measure costs

We consider the designed flood protection systems to have a lifetime of 90 years with revisions every 30 years depending on the discharge measurements made up to then. Since projections are only available until 2098 (viz. Sec. 3.3), we use the years 2009-2098 as the measure lifetime. At revisions, the flood protection may be adjusted (increases only). A GEV distribution with shape parameter $k$, time-dependent scale parameter $\beta(t) = \beta_0 + \beta_1 \times t$ and time-dependent location parameter $\mu = \mu_0 + \mu_1 \times t$ is used to model the annual maximum discharges. The model parameters $k, \beta_0, \beta_1, \mu_0, \mu_1$, which are learned from the projections (Sec. 3.3), are uncertain and hence represented by random variables.





The quantitative estimate of the hidden uncertainty was taken from (Dittes et al., 2017a). For the computation, we used 300 samples of annual maximum discharge in the period 1-30 years and 70 samples of annual maximum discharge in the period 31-60 years.

With an initial planning decision followed by two possible adjustments to the other possible protection systems (in years 30 and 60), the four flood protection systems introduced in Sect. 3.2 can result in the 16 strategies reproduced in Tab. 6 (adjustment to reduce protection is not considered). When a system is constructed at a later time (e.g. S3 is not implemented initially but first S1 and then the protection is increased to S3), the cost differs, and not just due to discounting. A new planning process has to be set up, topsoil has to be removed, and in the worst case, the entire measure has to be re-built (e.g. in the case of a flood protection wall whose statics that would not permit an extra meter in height). Constructing the polder however is an independent project and therefore independent of dike or wall heightening and timing. Thus, for the optimization minimizing the sum of construction costs and damage the following three costs are required:

- Cost of constructing the polder (this equals the cost difference of S1 to S2 and S3 to S4)
- Cost difference of S3 to S1 when S3 is chosen initially
- Cost difference of S3 to S1 when S1 is chosen initially and adjusted to S3 later.

The cost of S1 itself is not required as it acts as the baseline. However, it is known to lie at around €29 M.

In the given case study, these costs have to be estimated. The total construction cost of the polder is €55 M (RMD Consult, 2016). Since it protects the city of Kolbermoor (and various smaller cities) as well as Rosenheim, and in Sect. 3.5.4 it is estimated that Rosenheim suffered about 30 % of the losses of Rosenheim and Kolbermoor together, we here assume that Rosenheim would cover 30 % of the polder construction costs, i.e. €17 M. We estimate the cost difference of S3 to S1 when S3 is chosen initially to be €8 M. This is based on the presumption that there is a quasi-linear relationship between dike height and construction cost (Perosa, 2015) and the statement of the protection agency that €25 M are spent to increase capacity from 360 to 480 $m^3 s^{-1}$ (corresponding to roughly 3 m dike heightening). Finally, we estimate the cost difference of S3 to S1 when S1 is chosen initially and adjusted to S3 later to be €17 M. This is based on the planning authorities' statement that planning new protection walls (e.g. to fit on the top of dikes) would cost €1,500 per m length and planning is carried out for both riversides along a 5.7 km stretch of the river. The discounting rate is 2 %.

To return to the concept of flexibility (viz. Sect. 2.2 or for more details (Dittes et al., 2017b)), the decision to build the polder is a fully flexible one (it can be taken at any time at the same cost). The decision to heighten dikes and walls by 1 m would correspond to a flexibility parameter of 0.7 following (Dittes et al., 2017b), thus this can be considered as a partially flexible strategy.

Since the costs are rough estimates, we have run the optimization also with deviating values as a simple sensitivity analysis. Table 7 provides an overview of the different building cost scenarios considered for optimization.



## 4    Results

Here, we present and discuss results for the case study. In Sect. 4.1., damages for different peak discharges and damage functions are given. Planning recommendations and a sensitivity analysis are presented in Sect. 4.2.

### 4.1    Flood extent and damage assessment for selected flood events

For each of the four flood protection systems, we modeled six flood events with peak discharges varying from 520 to 740 $m^3 s^{-1}$. Based on the simulation results, we calculated the flood damages using the RAM and the SDAM by the Bavarian water management administration. Figure 8 and Tab. 8 summarize the estimated damage sums for the four protection systems depending on the peak discharge. Comparing the discharge-damage-curves, protection system 1 shows the greatest damages and thus offers the lowest flood protection of the four systems. The damage with S1 increases almost linearly over the peak discharge. Protection system 4 in contrast, results in the least damages of all protection systems. The discharge-damage-curves of S2 and S4 illustrate the effect of the polder. For the lower peak discharges, the polder reduces the flood wave to such an extent that no damage occurs up to a discharge of 610 $m^3 s^{-1}$. However, the damage increases significantly for S2 when the water overtops the dikes in the southeast of Rosenheim at 740 $m^3 s^{-1}$. The elevated protection system 3 shows damages at all peak discharges. These damage sums are however significantly lower than for S1 (Fig. 8, Tab. 8). The significant differences found among the damage models are consistent with the findings of the model verification (Sect. 3.5.4).

Figure 9 displays the simulated inundation extent and depths for S1 to S4 in case of a flood event with peak discharge 700 $m^3 s^{-1}$. With S1, the largest flooding occurs, which however can be drastically diminished by using the polder (S2). Compared to S1, the elevated protection system 2020 (S3) can reduce the flooded area and water depths significantly but not to the same extent as S2. With S4, flooding occurs only in and around the floodplain forest, which is not embanked for reasons of retention. In general, we note that the damage increases rather linearly with discharge for the protection systems without polder (S1 and S3). When using the retention volume of the polder, the flood peak is reduced, resulting in no or lower damage. Based on these results, we extrapolate linearly between damages for individual discharges as given in Fig. 8 and Tab. 8, using the resulting damage function in the optimization framework to arrive at the results given in the following section.

### 4.2    Risk assessment and planning recommendation

We show the result of the optimization, which is the system that is recommended for implementation at the present time, in Tab. 9. In order to evaluate robustness, three different damage models (RAM using the ATKIS dataset, RAM using the CORINE dataset and SDAM, viz. Sect. 3.5) were used, as well as different estimates of the required building cost (viz. Sect. 3.5). S4 – that is S3 plus the polder – is the recommended protection system in most cases. S3 – that is the further elevation of dikes and walls by 1 m in height in the course of the current building efforts – is recommended in the case of high or very high polder costs when the damage model is RAM ATKIS and in the case of very high polder costs also when the damage model is SDAM. In these cases, the polder may be built at a later time (corresponding to an upgrading to S4), which is considered in





the analysis. Strategies, in which a lower protection (S1 or S2) is built initially, with a possible extension to S3 or S4 later, are not optimal in any of the investigated cases.

We show the expected sum of life-time costs and risks in Tab. 10, expected life-time costs individually in Tab. 11 and expected life-time risks individually in Tab. 12, all based on implementation of the recommended protection system. In

Tab. 10 and Tab. 11, results are given for the different damage models and estimates of required building costs as in Tab. 9. However, results are not shown for differing costs of later elevation of the walls, since later elevation will not take place given that S4 has been recommended from the start. The life-time risks in Tab. 12 are independent of measure building costs yet dependent on the system that is initially implemented, hence they are shown for the different damage models and recommended protection system (rather than for the different damage models and estimates of required building costs). When just S1 is

implemented, the residual risk is €124 M according to the damage model that best fitted the damages of the 2013 flood, SDAM (viz. Sect. 3.5.4). Table 10-12 show that despite the higher associated risks, implementing S3 instead of S4 can be attractive due to the low building costs. Note that the results shown include the possible need for future adjustment of S3 to S4 (by constructing the polder). When using the SDAM damage model, the probability of later adjustment when S3 was recommended initially is 58%. With RAM using the ATKIS land cover, this probability is just 3% due to the very low damage estimates –

probably a strong underestimation, as discussed in Sect. 3.5.4.

In Fig. 10, we demonstrate how the need to adjust S3 to S4 might arise, using output from the case where S3 was recommended for initial implementation: damage model SDAM and very high polder costs. The decision is re-evaluated after 30 years, at which point it is decided whether the protection should remain unchanged or whether the polder should be constructed after all (i.e. S3 adjusted to S4). In panel (a), we give two examples of annual maximum discharges that may have

been observed during this first planning period: a set of relatively low discharges (orange dots) or a set of relatively high discharges (blue dots). For the former, no damages are incurred whereas for the latter, there are three floods. The damages caused by the floods are shown by the lilac bars. Depending on the discharges observed in the first planning period, the expected damage (risk) changes, as shown in panel (b). Initially, it was €30.1 M (petrol bar in year zero). After observing the first 30 years of discharges however, it changes to €48 M / €151 M (with / without adjustment to S4) in case of the high

discharges (yellow / petrol bar in year 30) and €20 M / €70 M (with / without adjustment to S4) in case of the low discharges (the latter is not shown). These numbers pertain to the then remaining lifetime (years 31-90) and are discounted to year 30. For the high discharges, the difference of adjustment to the expected damage is higher than the building cost of the polder and hence it is sensible to adjust.

So far, the protection recommendation has been given depending on measure costs and damage model. As has been

discussed in Sect. 3.5.4, RAM using the ATKIS land cover set likely leads to a significant underestimation of damages. Thus, the results return the robust recommendation to Rosenheim decision makers to choose the most conservative protection option, S4, unless they have to cover a strongly disproportionate share of the polder costs. We would recommend S4 even then, based on more qualitative arguments: polders have the benefit of providing a hierarchical (upstream as well as downstream)





protection (Custer and Nishijima, 2013) and are particularly robust with respect to changes in flood frequency, an aspect that is very desirable in protection planning (Baker et al., 2005; Merz et al., 2014). Additionally, heightening of dikes and walls is reaching a static and aesthetic limit in Rosenheim and thus if a polder can provide at least some of the necessary protection, it should be made use of.

According to the damage functions reproduced in Fig. 8, the risk-based recommendation for protection system S4 corresponds to a safety margin of 28 % with respect to the 100-year flood estimate, with moderate damages for discharges exceeding the protection. In a study for the same catchment and using largely the same methodology but aiming to protect against the 100-year flood (criterion-based rather than risk-based) and abstract protection levels rather than concrete measures, a safety margin of 12.5 % was recommended (Dittes et al., 2017a). The reason for this lies in the criterion-based optimization

neglecting damages. Since construction is dense in the endangered area, it is to be expected that the protection criterion should be higher than the 100-yr flood. This demonstrates that ignoring the damages caused by rare events can lead to economically sub-optimal protection recommendations.

## 5    Discussion

In this study, we investigated four flood protection systems with different safety levels for the city of Rosenheim. The basis

for the protection systems S1 and S2 were elaborated plans and concepts of technical protection measures. We further ensured the practicability and feasibility of the proposed flood protection systems through the close exchange with the local water management office. However, we can only assume that an elevation of the dikes and walls in Rosenheim by 1 m is feasible, as planned for S3 and S4. For a final confirmation, we would need expert opinions for every construction by engineering companies. Furthermore, it is unclear whether the protection measures work as intended in the event of an extreme flood.

Although most of the dikes in Rosenheim have an inner core (Fig. 5), a break of the homogeneous dikes in case of overtopping cannot be ruled out. In addition, the event-related control of the polder might cause problems. For example, if one or more floodgates are blocked or if the flow time or the volume of the flood wave are underestimated. In general, a technical or human failure is always possible.

     The recommendation for an initial flood protection system of Rosenheim results from a modeling sequence. The

uncertainty is handed over and increased from model to model: starting with the data inputs and the assumptions made, through to model and climate uncertainty. The uncertainty propagates from the hydrodynamic model through the damage model to the optimization framework. To reduce this uncertainty, we calibrated and validated our models using the available data. The hydrodynamic model was calibrated and validated solely based on recorded water levels of a flood event in 2001. We conducted a second qualitative model validation for the flood event of 2013 that brought us to the conclusion that the 2D-

model of Rosenheim simulates well. However, land use changes and sedimentation of the Mangfall River may alter the discharge in the future, which we did not consider in this study. Overall, uncertainty will always be part of such a study, and the analysis framework must address this uncertainty, as we do in our study.

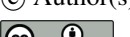



Compared to the hydrodynamic model, the validation of the damage models was more difficult due to the limited data available. As detailed damage data of the event 2013 were missing, the quality of the models could only be assessed by means of an approximate reported damage sum. It was shown that the RAM using ATKIS lead to a strong underestimation of replacement costs and hence RAM using the (coarser) CORINE land cover set is a better choice. We hypothesize that the

5 reason for this is the original calibration of the RAM to the coarser land use model, which inflates the built area compared to the fine 1 m x 1 m ATKIS model. Using the more detailed ATKIS dataset that, among others, correctly identifies river areas and applies no damage there, thus leads to an underestimation of the true damage. Overall, the variability of the damage estimates for the considered flood events (Fig. 8, Tab. 8) was high between the three damage approaches, indicating the significant uncertainty associated with loss estimation. In addition to the inaccuracy of the loss model, the future population

and value development plays a role in the assessment of losses, and is – at least for the further future – subject to large uncertainties as well.

The sequential nature of the decision process does not become relevant in this particular case study, since the most conservative strategy (S4) is recommended to be implemented initially. Thus, a static CBA would have yielded the same result here. However, this conclusion can only be drawn a posteriori: if a static CBA had been used from the start, one would not

know whether sequential planning may have led to a more optimal solution. When varying costs, strategy S3 was recommended initially for some cases (Tab. 9). These recommendations would likely be different when applying static CBA, as the proposed framework considers a high probability of adjustment to S4 at a later stage (58% in case of the best damage model).

Similarly, neglecting the 'hidden uncertainty' is unlikely to lead to a different recommendation in this case study – as the recommendation for S4 is very robust and, as shown in (Dittes et al., 2017a), including the hidden uncertainty in

Rosenheim increases the recommended planning margin only by a few percent – but again, this is knowledge of hindsight and may be very different in a different catchment. The spectrum of climate uncertainties will look different for other geographical locations (Hawkins and Sutton, 2009, 2011). In particular, forcing uncertainty – and thus the dependence on global socio-economic developments – may play a much larger role or the absolute amount of climate uncertainty may be larger than in the case study. Furthermore, the case study area is characterized by an exceptionally slow emergence of discharge trends (Maraun,

2013), thus almost anywhere else the trend will play a larger role.

It should be stressed that this paper aims to demonstrate how different sources of uncertainty can be combined to make robust decisions while taking into account future developments. To that end, the case study has an exemplary purpose rather than representing a definite recommendation for the study site. Instead, the tools presented here are intended to be used e.g. by climate scientists and hydrologists, which will have the care and expertise to include catchment-specific considerations.

In Rosenheim, one particular challenge for realistic recommendations lies in the discrepancy between historic record and projections: the projections exhibit a 100-year discharge that is 8 % higher in the historic time period than the 100-year discharge from the historic record. At first glance, this would suggest that one should use the historic record within the presented analysis to mitigate this discrepancy, rather than solely basing the optimization on projections. However, the



knowledge from historic discharge is implicit in the bias correction of the climate projections. Hence, 'ideal' input projections would not exhibit a systematic discrepancy to historic data and performing some kind of post-correction within the framework may be a double-correction. Further considerations that speak against a post-correction in the framework are that the short length of the historic record implies a large uncertainty, that the correction may have compromised the model spread and that

potentially valuable regional information is contained in the projections (which have been calibrated to the Inn valley). It is clear from the Rosenheim data and the literature, however, that there is a considerable need for projections and bias correction methods with a focus on extreme values.

Finally, it should be kept in mind that damage potential and protection are not independent in reality. Instead, an increase in flood protection may encourage settlement patterns that increase the damage potential, e.g. houses would not be

built in an unprotected flood plain yet they are built behind dikes protecting against moderate floods, leading to strongly increased losses in the case of large floods (IPCC, 2012; Seifert, 2012). It is a challenge for authorities to, at the same time, justify the construction of unpopular technical flood protection to their citizens and restrict building permissions to break this vicious cycle. This leads us to a more general discussion on the relative virtues of risk-based planning – which considers the damage potential – and criterion-based planning – which protects areas from floods of a fixed return period. In an area with

low damage potential, protecting from the 30-year flood may be sufficient, and any investments in protection that goes beyond this may not be economically sensible. In an area with high damage potential, it may be economically sensible to protect also from much rarer floods. The latter appears to be the case in the case study area of Rosenheim. Whether it is under- or overprotection, a fixed protection criterion will typically lead to sub-optimal results. Therefore, the trend in flood protection planning is towards a risk-based approach, as has manifested itself in the European flood risk directive 2007/60 (European

Parliament and European Council, 2007). Nevertheless, criterion-based flood protection may at times make sense as a measure of public planning. Much of the damage potential will be from private buildings and it is arguably not the responsibility of the state to protect the full asset value. Deciding to provide protection from the 100-year flood to all can be a fair solution, in that tax money is not disproportionately spend on those owning, or deciding to build, high-asset properties in the floodplain. Giving some responsibility – e.g. to add local protection measures or insurance at their own expense – to the citizens concerned also

curbs the mentioned 'feedback loop' of flood protection planning. Another issue with risk-based planning is, that it is often not clear how to calculate the damage potential: should it include only public buildings and critical infrastructure? All buildings? The costs of downtime of local industry? Should the benefits to the economy from reconstruction efforts be deducted? And should the appraisal be for replacement costs or depreciated value? Furthermore, based on lack of data, it is often much more complicated to estimate damage potential than flood frequency. Simply protecting from a design flood of

fixed return period avoids these issues and, potentially, the discussions or even lawsuits that come with them. Thereby, it also allows for faster planning processes. However, when there are the resources to do so, we would always recommend that planning agencies do at least a simple risk-based evaluation as part of their planning process, to avoid gross sub-optimality.



## 6    Conclusions

We have conducted a risk-based evaluation of four alternative flood protection systems for the pre-alpine city of Rosenheim in the Mangfall catchment, Bavaria, Germany. To do so, several damage model and building cost estimates have been used in a fully quantitative Bayesian optimization framework taking into account climate uncertainty and the possibility to adjust the

measures at a later time. The recommendation is robustly for the most conservative strategy, which includes a further heightening of dikes and walls by 1 m over the 100-year protection including freeboard as well as a large upstream polder. This recommendation is in contrast to the less conservative recommendation obtained when following a sequential planning that aims at compliance with the minimum protection level, which does not require an assessment of damages. Thus, the case study underlines the importance of taking a risk-based approach in flood protection planning. It also becomes clear that even

when there is a large uncertainty in damage, costs, and climatic development, there need still not be ambiguity about the protection decision.

## Code availability

The code is available upon request.

## Data availability

The projections of annual maximum discharges used are available in the supplement.

## Author contribution

This work was conducted by B. Dittes and M. Kaiser under the supervision and guidance of O. Špačková, W. Rieger, M. Disse and D. Straub. M. Kaiser prepared Sect. 3.1-3.2, 3.4-3.5 and 4.1 as well as the respective parts of the discussion and performed the analyses described in her sections. B. Dittes prepared the remainder of the paper and performed the remaining analyses.

All authors contributed to the final writing of the paper.

## Competing interests

The authors declare that they have no conflict of interest.

## Acknowledgements

We would like to thank Bayerisches Landesamt für Umwelt (LfU), Wasserwirtschaftsamt Rosenheim (WWA-Ro) and

Bayerisches Staatsministerium für Umwelt und Verbraucherschutz (StMUV) for fruitful discussions. The LfU also provided



the discharge records and projections used in the case study. Most projections were modelled within the cooperation KLIWA. These are based on ENSEMBLES data funded by the EU FP6 Integrated Project ENSEMBLES (Contract number 505539), whose support is gratefully acknowledged. The remaining projections are REMO1 ('UBA') and REMO2 ('BfG') (Umweltbundesamt, 2017), as well as CLM1 and CLM2 (Hollweg et al., 2008). The WWA-Ro provided the hydrodynamic

model of the study site. This work was supported by Deutsche Forschungsgemeinschaft (DFG) through the TUM International Graduate School of Science and Engineering (IGSSE). The ICPR kindly provided us with the RAM as an executable ArcGIS toolbox Finally, we thank Adrian Schmid-Breton from the International Commission for the Protection of the Rhine for providing the Rhine Atlas model and his kind support.

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



**Figures**

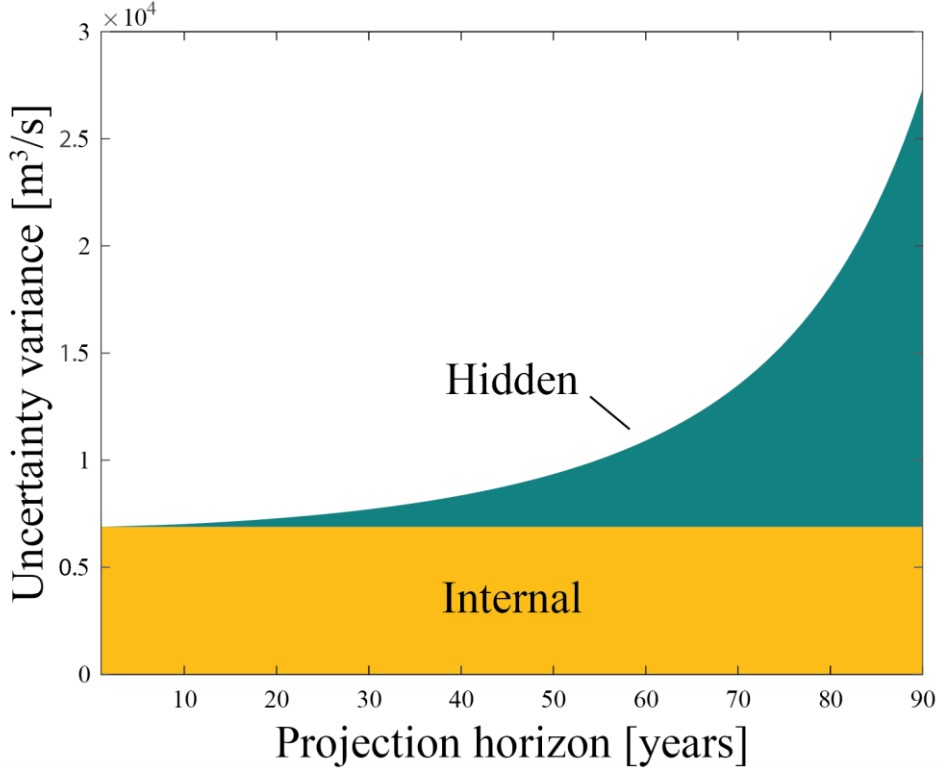

**Figure 1.** Absolute values of hidden uncertainty and internal variability over the projection horizon for the CCLM projection.





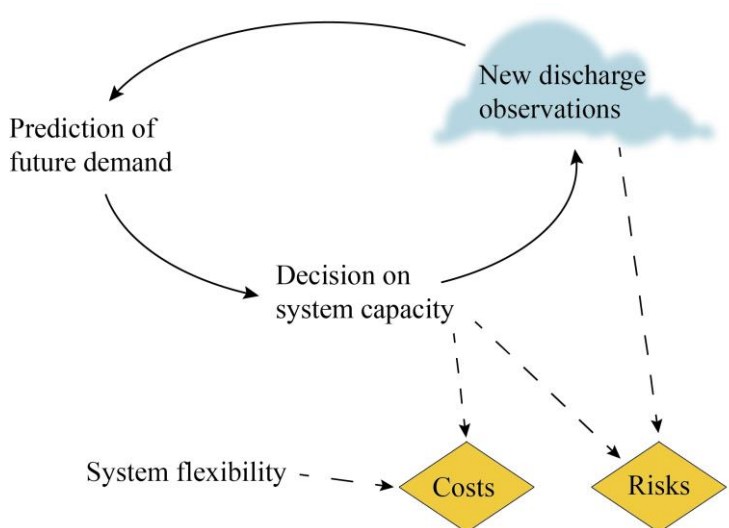

**Figure 2.** Schematic sequential planning process: after an initial decision on system capacity, new observations lead to an altered prediction of future demand and thus potentially adjustment. At the same time, the future discharges cause damages (i.e., in there expected form, risks) depending on the capacity in place. The cloud signifies that future discharges are uncertain. Costs of decisions depend on system flexibility. Adapted from (Špačková et al., 2015).

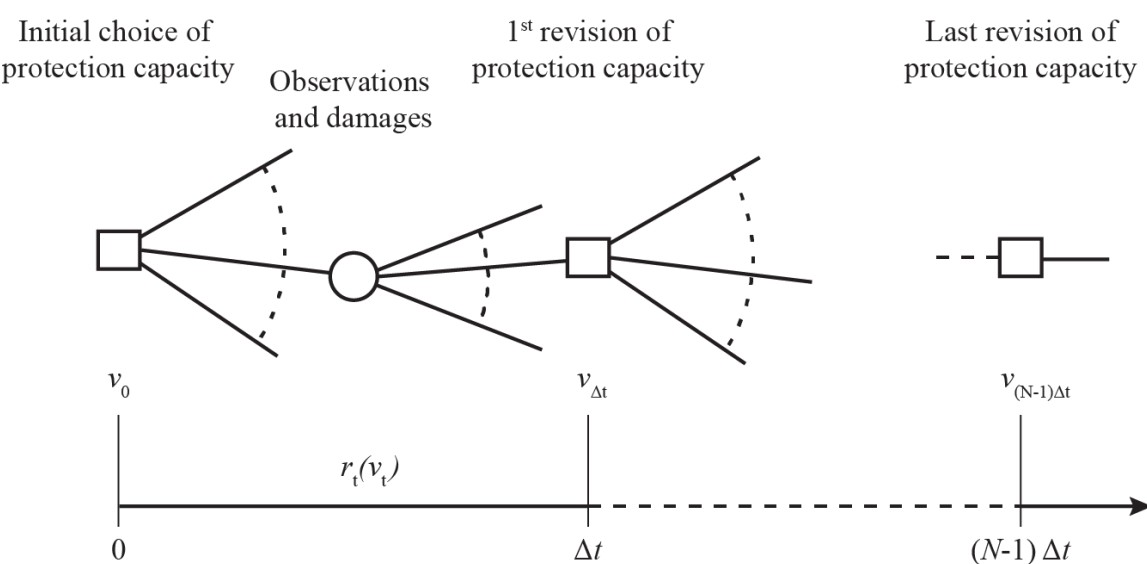

**Figure 3.** Sequential planning process in risk-based flood protection planning. Squares stand for protection decisions and circles for observations of annual maximum discharge. Adapted from (Dittes et al., 2017b).



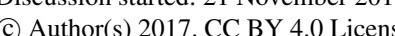

**Figure 4.** The Mangfall catchment and the case study site Rosenheim, Germany (Data source: Geobasisdaten © Bayerische
Vermessungsverwaltung, www.geodaten.bayern.de).



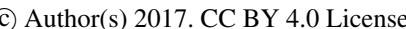

**Figure 5.** Protection system 2020 (S1) along the Mangfall River and the creeks Auerbach and Kaltenbach in the municipal area of Rosenheim

(Data source: Geobasisdaten © Bayerische Vermessungsverwaltung, www.geodaten.bayern.de)




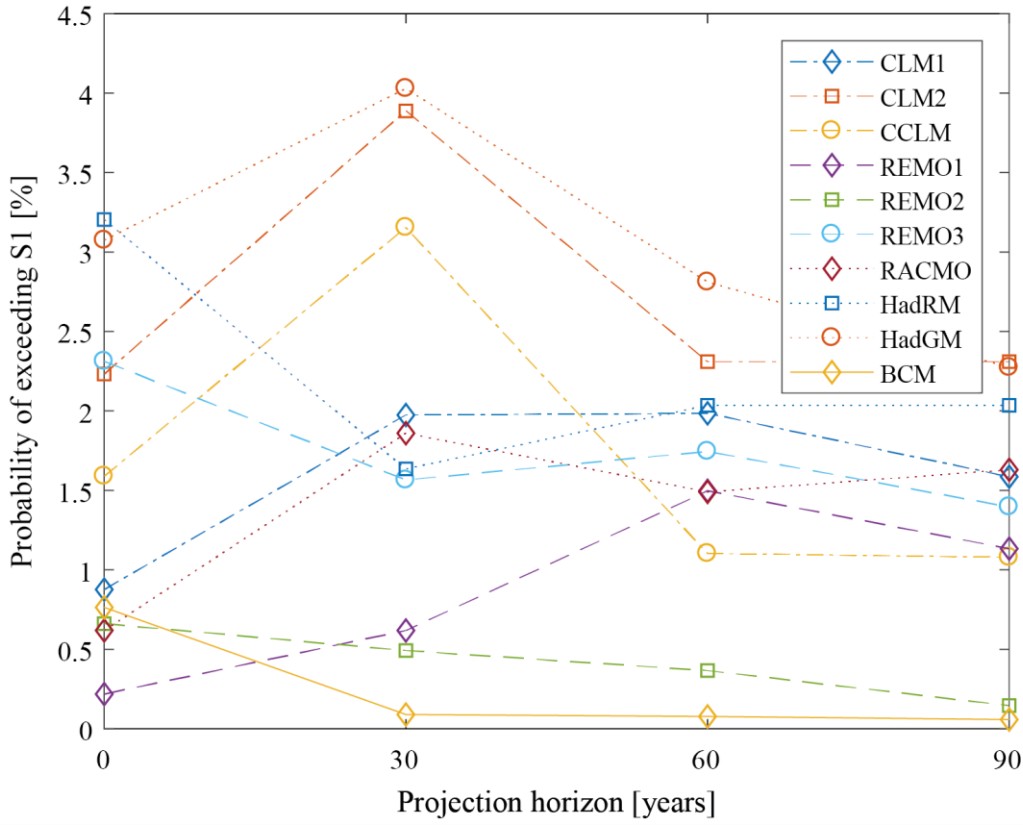

**Figure 6.** Probability of exceeding in S1 (480 m³s⁻¹) for the individual projections at initial planning (year 0) and at later time points (years 30, 60 and 90).





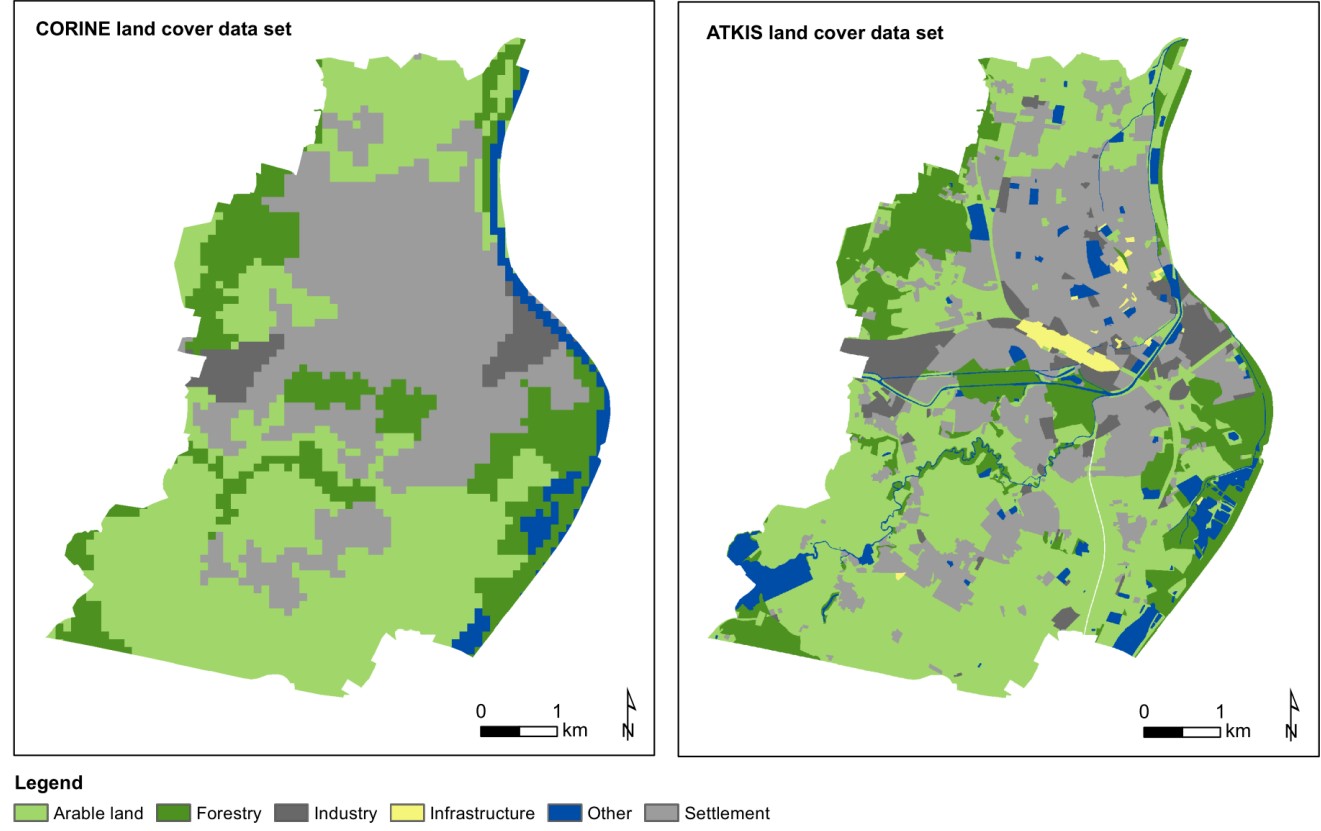

**Figure 7.** Classification of the study site into the damage categories of the Rhine Atlas model based on the CORINE and ATKIS land cover data set (CLC 2006, ATKIS®-Basis-DLM 2008)





**Figure 8.** Estimated flood damages for four protection systems and varying peak discharges. Flood damages were calculated using the Rhine Atlas model (RAM) with different land use data sets and a simple damage assessment method (SDAM) by the Bavarian water management administration.





**Figure 9.** Modeled inundation depths for a flood event of 700 m$^3$ s$^{-1}$ and four different protection systems (Data source: Geobasisdaten ©

Bayerische Vermessungsverwaltung, www.geodaten.bayern.de)





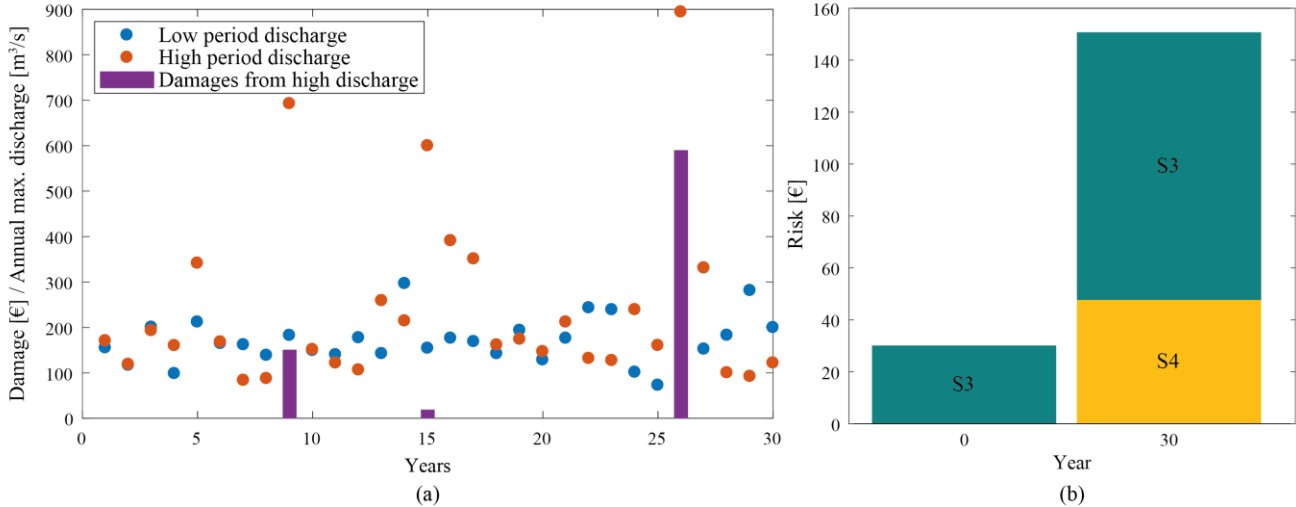

**Figure 10.** Example of changing risk estimate due to future observations. (a) Two different realizations of annual maximum discharges in the first period (year 1-30) and respective flood damages (lilac bars) when system S3 is implemented initially. (b) Expected future damage (i.e. risk) for the protection system life-time when S3 is implemented initially (year 0) and after 30 years for the remaining life-time in case the high period discharge was observed with (yellow) / without (petrol) adjusting to S4.





## Tables

**Table 1.** Regional climate models (RCMs) used in this study, driving global climate models (GCMs), source of the RCMs, downscaling and hydrological model.

| Name | GCM | RCM | Source | Downscaling | Hydrological model |
|---|---|---|---|---|---|
| CLM1 | ECHAM5 R1 | CLM Consortial | Consortium | Quantile mapping (German federal institute of hydrology BfG), SCALMET (Willems and Stricker, 2011) | |
| CLM2 | ECHAM5 R2 | CLM Consortial | Consortium | | |
| CCLM | HadCM3Q0 | CCLM | ETH | | |
| REMO1 | ECHAM5 R1 | REMO | MPI | | |
| REMO2 | ECHAM5 R2 | REMO | MPI | | WaSiM v8.06.02, Inn, daily, 1 km² |
| REMO3 | ECHAM5 R3 | REMO | MPI | | |
| RACMO | ECHAM5 R3 | RACMO2 | KNMI | Quantile mapping (Bavarian Environment Agency LfU), SCALMET (Schmid et al., 2014) | |
| HadRM | HadCM3Q3 | HadRM3Q3 | Hadley Centre | | |
| HadGM | HadCM3Q3 | RCA3 | SMHI | | |
| BCM | BCM | RCA3 | SMHI | | |



**Table 2.** Adaptation of the specific asset values stored in the Rhine Atlas model for Bavaria to the case study site Rosenheim

| | | Specific asset value per damage category [€ m$^{-2}$] | | | | | |
|---|---|---|---|---|---|---|---|
| | | Settlement | Industry | Infrastructure | Arable land | Forestry | Other |
| Bavaria | immobile | 300.00 | 294.00 | 276.00 | 7.00 | 2.00 | - |
| | mobile | 61.00 | 93.00 | 2.00 | - | - | - |
| Rosenheim | immobile | 328.80 | 322.22 | 302.50 | 7.67 | 2.19 | - |
| | mobile | 66.86 | 101.93 | 2.19 | - | - | - |





**Table 3.** Assignment of the CORINE and ATKIS land cover classes of Rosenheim to the damage categories of the Rhine Atlas model (RAM) (CLC 2006, ATKIS®-Basis-DLM 2008)

| Damage category of RAM | CORINE land cover class | CLC Code | Area [km²] | Percentage of area [%] | ATKIS object type | ATKIS Code | Area [km²] | Percentage of area [%] |
|---|---|---|---|---|---|---|---|---|
| Settlement | Continuous urban fabrics | 111 | 12.7 | 34 | Residential area | 2111 | 10.3 | 28 |
| | Discontinuous urban fabric | 112 | | | Area of mixed use | 2113 | | |
| Industry | Industrial or commercial units | 121 | 1.2 | 3 | Industrial or commercial units | 2112 | 2.2 | 6 |
| Infrastructure | - | - | - | - | Square | 3103 | 0.4 | 1 |
| | | | | | Central station | 3501 | | |
| Arable land | Non-irrigated arable land | 211 | 15.9 | 43 | Arable land | 4101 | 16.3 | 44 |
| | Pastures | 231 | | | Greenland | 4102 | | |
| | Complex cultivation patterns | 242 | | | Gardenland | 4103 | | |
| | Land partially occupied by agriculture | 243 | | | Specialized cultivation | 4109 | | |
| Forestry | Broad-leaved forest | 311 | 6.5 | 17 | Forest | 4107 | 5.7 | 15 |
| | Coniferous forest | 312 | | | Wood | 4108 | | |
| | Mixed forest | 313 | | | | | | |
| Other | Water courses | 511 | 1.1 | 3 | Sports facility | 2201 | 2.3 | 6 |
| | Water bodies | 512 | | | Leisure facility | 2202 | | |
| | | | | | Cemetery | 2213 | | |
| | | | | | Mine, stone pit | 2301 | | |
| | | | | | Moorland | 4104 | | |
| | | | | | Moor, moss | 4105 | | |
| | | | | | River, stream | 5101 | | |
| | | | | | Trench, canal | 5103 | | |
| | | | | | Lake, pond | 5112 | | |
| | | | | | Protected landscape | 7302 | | |
| | | Sum | 37.3 | 100 | | Sum | 37.2 | 100 |



**Table 4.** Usage factors by building category according to the water management office of Rosenheim

| Building category | Usage factor $f_u$ |
|---|---|
| Normal house up to 2 apartments | 1 |
| Apartment building more than 3 apartments | 3 |
| Small-sized business, service provider (chancellery, practice): up to 3 jobs | 2 |
| Medium-sized business, service provider (chancellery, practice): 4 to 49 jobs (interpolate linearly) | 2–10 |
| Large-sized business (more than 49 jobs) | 10 |
| Public institution (hospital, retirement home, school, etc.) | 10 |
| Important infrastructure (water supply, power station) | 10 |
| Special cases must be considered separately | |

*in case of adjoining building:* $f_u \cdot 0.5$



**Table 5.** Comparison of loss estimates with the reported loss range for the flood event of June 2013 in Rosenheim

| Damage model | Damage estimate [M €] |
|---|---|
| Rhine Atlas model | |
|     CORINE land cover | 97 |
|     ATKIS land cover | 15 |
| Simple damage estimation method | 78 |
| Reported loss range (replacement costs) | 38–70 |

5   **Table 6.** Potential protection strategies for Rosenheim implemented over the measure life time. Protection system S1 corresponds to the current protection whereas system S4 corresponds to the current protection plus 1 m heightening of dikes and walls plus a flood polder (retention basin). Lowering the protection is not considered.

| Initial decision | S1 | S1 | S1 | S1 | S1 | S1 | S1 | S1 | S1 | S2 | S2 | S2 | S3 | S3 | S3 | S4 |
|---|---|---|---|---|---|---|---|---|---|---|---|---|---|---|---|---|
| Revision I (at 30 years) | S1 | S1 | S1 | S1 | S2 | S2 | S3 | S3 | S4 | S2 | S2 | S4 | S3 | S3 | S4 | S4 |
| Revision II (at 60 years) | S1 | S2 | S3 | S4 | S2 | S4 | S3 | S4 | S4 | S2 | S4 | S4 | S3 | S4 | S4 | S4 |





**Table 7.** Cost estimates used for optimization. In order to study sensitivity, polder costs, the costs of increasing dikes / walls initially and the costs of increasing dikes/walls later were varied.

| Name | Measure costs [M €] | | |
| --- | --- | --- | --- |
| | Polder | Add 1 m height initially | Add 1 m height later |
| Reference | 17 | 8 | 17 |
| Higher polder costs | 30 | 8 | 17 |
| Very high polder costs | 55 | 8 | 17 |
| Higher costs 1 m initially | 17 | 12 | 17 |
| Very high costs 1 m initially | 17 | 15 | 17 |
| Lower costs 1 m later | 17 | 8 | 8 |
| Very low costs 1 m later | 17 | 8 | 5 |





**Table 8.** Damages estimated using three different flood models depending on protection system in place.

| Protection system | Discharge [m³ s⁻¹] | Damage model [M €] | | |
| --- | --- | --- | --- | --- |
| | | RAM ATKIS | RAM CORINE | SDAM |
| S1 | 518 | 3.9 | 90 | 10 |
| | 584 | 60 | 170 | 210 |
| | 614 | 120 | 230 | 280 |
| | 652 | 140 | 260 | 310 |
| | 698 | 160 | 290 | 360 |
| | 743 | 240 | 400 | 540 |
| S2 | 518 | 0 | 0 | 0 |
| | 584 | 0 | 0 | 0 |
| | 614 | 0 | 0 | 0 |
| | 652 | 10 | 90 | 40 |
| | 698 | 20 | 90 | 60 |
| | 743 | 150 | 270 | 290 |
| S3 | 518 | 2.5 | 90 | 10 |
| | 584 | 4.1 | 100 | 10 |
| | 614 | 10 | 110 | 40 |
| | 652 | 20 | 130 | 110 |
| | 698 | 40 | 160 | 160 |
| | 743 | 100 | 240 | 270 |
| S4 | 518 | 0 | 0 | 0 |
| | 584 | 0 | 0 | 0 |
| | 614 | 0 | 0 | 0 |
| | 652 | 2.4 | 90 | 4.2 |
| | 698 | 2.7 | 90 | 10 |
| | 743 | 10 | 110 | 60 |





**Table 9.** Initial protection system recommended by optimization framework. In the case of higher polder costs and SDAM, the recommendation differs between projection sets.

| Build costs \ Damage model | RAM ATKIS | RAM CLC | SDAM |
|---|---|---|---|
| Reference | S4 | S4 | S4 |
| Higher polder costs | S3 | S4 | S4 |
| Very high polder costs | S3 | S4 | S3 |
| Higher costs 1m initially | S4 | S4 | S4 |
| Very high costs 1m initially | S4 | S4 | S4 |
| Lower costs 1m later | S4 | S4 | S4 |
| Very low costs 1m later | S4 | S4 | S4 |

**Table 10.** Life-time costs + risks (sum) [M €]

| Build costs \ Damage model | RAM ATKIS | RAM CLC | SDAM |
|---|---|---|---|
| Reference | 27.8 | 47.8 | 42.6 |
| Higher polder costs | 32.0 | 60.8 | 55.6 |
| Very high polder costs | 32.7 | 85.8 | 70.2 |
| Higher costs 1m initially | 31.8 | 51.8 | 46.6 |
| Very high costs 1m initially | 34.8 | 54.8 | 49.6 |



**Table 11.** Life-time costs [M €]

| Build costs \ Damage model | RAM ATKIS | RAM CLC | SDAM |
|---|---|---|---|
| Reference | 25.0 | 25.0 | 25.0 |
| Higher polder costs | 8.8 | 38.0 | 38.0 |
| Very high polder costs | 9.5 | 63.0 | 40.1 |
| Higher costs 1m initially | 29.0 | 29.0 | 29.0 |
| Very high costs 1m initially | 32.0 | 32.0 | 32.0 |

**Table 12.** Life-time risks [M €]

| Initial system \ Damage model | RAM ATKIS | RAM CLC | SDAM |
|---|---|---|---|
| S3 | 23.2 | - | 30.1 |
| S4 | 2.8 | 22.8 | 17.6 |

# A      Discharge projections at Rosenheim

**Table A 1.** Annual maximum discharges [m$^3$ s$^{-1}$] projected at Rosenheim ($p$) during the planning horizon of 90 years, based on WaSiM v8.06.02, Inn, daily, 1 km².

| Year | CLM1 | CLM2 | CCLM | REMO1 | REMO2 | REMO3 | RACMO | HadRM | HadGM | BCM |
|---|---|---|---|---|---|---|---|---|---|---|
| 2009 | 148 | 117 | 313 | 165 | 83.9 | 237 | 393 | 205 | 216 | 164 |
| 2010 | 107 | 166 | 258 | 154 | 113 | 241 | 225 | 140 | 264 | 277 |
| 2011 | 140 | 138 | 236 | 123 | 166 | 148 | 127 | 126 | 164 | 160 |
| 2012 | 269 | 168 | 246 | 194 | 136 | 258 | 110 | 212 | 268 | 352 |
| 2013 | 113 | 171 | 196 | 195 | 170 | 145 | 146 | 196 | 200 | 249 |
| 2014 | 131 | 102 | 252 | 134 | 190 | 105 | 188 | 253 | 194 | 214 |
| 2015 | 216 | 238 | 251 | 174 | 208 | 111 | 135 | 239 | 284 | 171 |
| 2016 | 181 | 244 | 128 | 241 | 264 | 331 | 659 | 98.5 | 117 | 129 |
| 2017 | 134 | 274 | 178 | 139 | 306 | 128 | 166 | 291 | 275 | 131 |
| 2018 | 457 | 294 | 370 | 609 | 233 | 183 | 140 | 245 | 132 | 168 |
| 2019 | 219 | 116 | 202 | 282 | 133 | 156 | 181 | 138 | 145 | 228 |



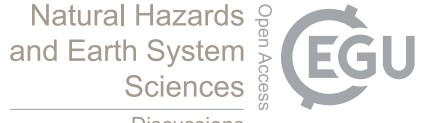

| | | | | | | | | | |
|---|---|---|---|---|---|---|---|---|---|
| 2020 | 185 | 206 | 336 | 232 | 178 | 212 | 367 | 139 | 177 | 161 |
| 2021 | 107 | 163 | 176 | 91 | 221 | 171 | 90.8 | 172 | 159 | 309 |
| 2022 | 140 | 431 | 189 | 322 | 226 | 104 | 155 | 518 | 193 | 180 |
| 2023 | 418 | 270 | 202 | 225 | 147 | 227 | 332 | 169 | 146 | 144 |
| 2024 | 216 | 126 | 283 | 95 | 173 | 158 | 124 | 215 | 168 | 218 |
| 2025 | 131 | 130 | 165 | 184 | 116 | 225 | 132 | 141 | 215 | 246 |
| 2026 | 149 | 183 | 403 | 135 | 125 | 328 | 156 | 116 | 149 | 195 |
| 2027 | 152 | 201 | 452 | 174 | 250 | 158 | 170 | 157 | 335 | 189 |
| 2028 | 251 | 371 | 146 | 197 | 425 | 138 | 182 | 390 | 552 | 371 |
| 2029 | 181 | 238 | 369 | 153 | 205 | 424 | 230 | 175 | 323 | 169 |
| 2030 | 552 | 225 | 188 | 238 | 161 | 102 | 140 | 168 | 397 | 217 |
| 2031 | 175 | 370 | 399 | 175 | 104 | 101 | 125 | 159 | 381 | 90.1 |
| 2032 | 157 | 533 | 181 | 105 | 250 | 131 | 179 | 138 | 424 | 224 |
| 2033 | 135 | 213 | 67.2 | 319 | 183 | 193 | 168 | 133 | 185 | 166 |
| 2034 | 277 | 110 | 207 | 123 | 234 | 291 | 477 | 100 | 417 | 207 |
| 2035 | 253 | 118 | 333 | 134 | 147 | 165 | 310 | 281 | 142 | 214 |
| 2036 | 284 | 156 | 249 | 159 | 172 | 108 | 58 | 191 | 282 | 178 |
| 2037 | 185 | 155 | 92 | 94.8 | 140 | 156 | 125 | 157 | 121 | 165 |
| 2038 | 270 | 123 | 197 | 241 | 207 | 493 | 137 | 126 | 253 | 343 |
| 2039 | 196 | 237 | 193 | 173 | 180 | 130 | 142 | 134 | 225 | 175 |
| 2040 | 238 | 163 | 211 | 123 | 232 | 332 | 215 | 199 | 210 | 141 |
| 2041 | 166 | 116 | 194 | 145 | 140 | 240 | 171 | 229 | 192 | 308 |
| 2042 | 158 | 139 | 144 | 206 | 180 | 102 | 165 | 239 | 149 | 154 |
| 2043 | 365 | 139 | 90.7 | 377 | 160 | 110 | 78.2 | 182 | 76.9 | 107 |
| 2044 | 216 | 321 | 280 | 145 | 126 | 110 | 220 | 162 | 179 | 180 |
| 2045 | 188 | 186 | 301 | 196 | 165 | 150 | 118 | 188 | 137 | 315 |
| 2046 | 255 | 591 | 219 | 158 | 170 | 130 | 297 | 425 | 449 | 195 |
| 2047 | 141 | 191 | 331 | 146 | 291 | 229 | 192 | 160 | 328 | 219 |
| 2048 | 95.5 | 332 | 164 | 96.3 | 265 | 69.9 | 92.9 | 182 | 298 | 237 |
| 2049 | 194 | 176 | 226 | 199 | 214 | 170 | 133 | 151 | 124 | 235 |
| 2050 | 388 | 412 | 240 | 154 | 296 | 118 | 190 | 181 | 125 | 189 |
| 2051 | 103 | 92.3 | 140 | 112 | 154 | 209 | 212 | 276 | 237 | 239 |
| 2052 | 367 | 128 | 153 | 423 | 58.8 | 85.2 | 271 | 141 | 181 | 163 |
| 2053 | 196 | 263 | 303 | 157 | 91.1 | 173 | 207 | 64.6 | 135 | 137 |
| 2054 | 343 | 112 | 202 | 155 | 122 | 347 | 169 | 167 | 263 | 146 |
| 2055 | 99.4 | 159 | 142 | 172 | 266 | 160 | 114 | 241 | 229 | 101 |
| 2056 | 156 | 111 | 261 | 337 | 147 | 138 | 166 | 90.5 | 131 | 198 |
| 2057 | 152 | 278 | 123 | 128 | 175 | 194 | 269 | 138 | 214 | 148 |
| 2058 | 132 | 208 | 249 | 142 | 180 | 201 | 180 | 472 | 220 | 142 |

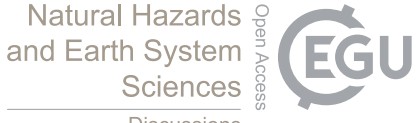



| | | | | | | | | | |
|---|---|---|---|---|---|---|---|---|---|
| 2059 | 157 | 287 | 178 | 208 | 316 | 271 | 288 | 190 | 163 | 235 |
| 2060 | 152 | 148 | 124 | 81.2 | 171 | 208 | 158 | 117 | 346 | 311 |
| 2061 | 381 | 299 | 290 | 72.1 | 245 | 186 | 79.1 | 115 | 305 | 161 |
| 2062 | 134 | 176 | 170 | 89.5 | 220 | 300 | 190 | 223 | 187 | 124 |
| 2063 | 160 | 150 | 143 | 217 | 135 | 281 | 401 | 119 | 236 | 325 |
| 2064 | 156 | 186 | 342 | 94.2 | 208 | 112 | 145 | 158 | 90.6 | 192 |
| 2065 | 353 | 104 | 213 | 170 | 273 | 142 | 202 | 189 | 170 | 175 |
| 2066 | 210 | 65.4 | 268 | 110 | 238 | 130 | 119 | 171 | 170 | 195 |
| 2067 | 628 | 145 | 91.5 | 195 | 168 | 162 | 401 | 147 | 141 | 265 |
| 2068 | 202 | 208 | 102 | 106 | 357 | 175 | 339 | 210 | 238 | 150 |
| 2069 | 285 | 116 | 309 | 146 | 59.5 | 154 | 321 | 186 | 254 | 249 |
| 2070 | 94.9 | 208 | 278 | 64.2 | 105 | 169 | 123 | 284 | 156 | 174 |
| 2071 | 278 | 132 | 350 | 188 | 72 | 217 | 154 | 248 | 239 | 171 |
| 2072 | 204 | 119 | 151 | 95.3 | 196 | 121 | 206 | 175 | 222 | 243 |
| 2073 | 122 | 157 | 216 | 184 | 201 | 175 | 227 | 86.1 | 332 | 259 |
| 2074 | 284 | 218 | 207 | 200 | 126 | 159 | 235 | 177 | 172 | 138 |
| 2075 | 148 | 78.6 | 96.1 | 133 | 97.9 | 118 | 281 | 184 | 170 | 145 |
| 2076 | 247 | 277 | 238 | 123 | 140 | 123 | 305 | 243 | 206 | 169 |
| 2077 | 116 | 216 | 131 | 101 | 229 | 231 | 217 | 121 | 192 | 107 |
| 2078 | 130 | 192 | 325 | 153 | 183 | 203 | 225 | 181 | 275 | 158 |
| 2079 | 155 | 101 | 161 | 232 | 152 | 193 | 95.5 | 127 | 170 | 141 |
| 2080 | 319 | 91.4 | 97.7 | 155 | 120 | 94.4 | 187 | 175 | 172 | 241 |
| 2081 | 257 | 194 | 270 | 253 | 158 | 190 | 206 | 183 | 262 | 192 |
| 2082 | 243 | 225 | 191 | 230 | 123 | 123 | 191 | 121 | 124 | 230 |
| 2083 | 110 | 106 | 128 | 87.6 | 160 | 192 | 171 | 293 | 295 | 213 |
| 2084 | 180 | 183 | 200 | 217 | 276 | 305 | 261 | 197 | 114 | 249 |
| 2085 | 92.3 | 223 | 194 | 80.4 | 134 | 216 | 308 | 288 | 143 | 190 |
| 2086 | 188 | 299 | 148 | 89.1 | 121 | 130 | 301 | 304 | 317 | 431 |
| 2087 | 208 | 189 | 309 | 111 | 106 | 147 | 342 | 82.9 | 250 | 179 |
| 2088 | 135 | 151 | 277 | 194 | 145 | 93.2 | 137 | 85.4 | 149 | 257 |
| 2089 | 213 | 135 | 119 | 186 | 122 | 90.7 | 409 | 121 | 141 | 89.2 |
| 2090 | 143 | 290 | 219 | 248 | 130 | 174 | 195 | 159 | 229 | 177 |
| 2091 | 57.6 | 188 | 295 | 80.6 | 152 | 132 | 267 | 181 | 253 | 166 |
| 2092 | 562 | 240 | 179 | 185 | 168 | 346 | 830 | 125 | 150 | 245 |
| 2093 | 154 | 227 | 355 | 94.1 | 187 | 148 | 175 | 178 | 199 | 166 |
| 2094 | 150 | 113 | 106 | 85 | 128 | 82 | 148 | 111 | 566 | 243 |
| 2095 | 185 | 187 | 340 | 111 | 153 | 145 | 163 | 158 | 143 | 174 |
| 2096 | 244 | 119 | 208 | 115 | 91.1 | 298 | 211 | 165 | 158 | 79.2 |
| 2097 | 306 | 133 | 74.3 | 267 | 121 | 221 | 170 | 610 | 187 | 200 |



| 2098 | 93.2 | 122 | 138 | 71.6 | 110 | 214 | 239 | 240 | 218 | 115 |