# Peer review of "Risk-based flood protection planning under climate change and modelling uncertainty: a pre-alpine case study"

_Natural Hazards and Earth System Sciences, 2017_

## Referee Comment (RC1) · Anonymous Referee #1 · 15 Jan 2018

The paper 'Risk based flood protection planning under climate change and modelling uncertainty: a pre-alpine case study' by B. Dittes et al. applies a framework for quantitative, probabilistic flood protection planning to a real decision making problem of flood protection strategies. This framework considers climatic uncertainties by incorporating non-stationarity and accounts for flexibility of the flood protection system in a sequential Bayesian approach. The planning problem investigated considers four alternative protection strategies with different safety levels for a city in a pre-alpine catchment. The topic is of high relevance in the context of adaptation planning and risk based decision making under uncertainty. The paper is very well written and structured. It will surely make an important contribution to the field. However, some aspects need further

consideration and explanation. The most important ones are:

The rationale and details of the approach to determine future extreme discharges is not described comprehensively and is hard to follow at times. Reference is made to the paper Dittes et al. 2017a which is however still under review by another Journal. It would be very helpful to provide more details on the background, e.g. how the standard deviation for the hidden uncertainty is quantified, what are the underlying assumptions.

The approach of backwards induction optimization (page 5) should be introduced more in detail. Particularly, the context that system performance is evaluated by taking data into account which is available by some point in the future needs some additional explanation. In your study you use discharges based on climate projections which are available today. The actually observed discharges in the future may differ from these projections, and thus may give different results and recommendations. Is this reflected by the uncertainty range of discharge projections from the climate scenarios?

The whole paragraph (page 14, ll 3-15) is not very clear. It includes a number of statements for which the basis is not comprehensible.

Your discussion of how the damage potential p 17 ll 23 - 30 should be extended by taking the broader perspective of costs of natural hazards introduced by Kreibich et al. 2014 into consideration. (Kreibich, H., van den Bergh, J. C. J. M., Bouwer, L. M., Bubeck, P., Ciavola, P., Green, C., Hallegatte, S., Logar, I., Meyer, V., Schwarze, R. and Thieken, A. H.: Costing natural hazards, Nature Clim. Change, 4(5), 303–306, doi:10.1038/nclimate2182, 2014.)

Additional minor comments and suggestions are included in the The paper 'Risk based flood protection planning under climate change and modelling uncertainty: a pre-alpine case study' by B. Dittes et al. applies a framework for quantitative, probabilistic flood protection planning to a real decision making problem of flood protection strategies. This framework considers climatic uncertainties by incorporating non-stationarity and accounts for flexibility of the flood protection system in a sequential Bayesian approach.

[Figure]

The planning problem investigated considers four alternative protection strategies with different safety levels for a city in a pre-alpine catchment. The topic is of high relevance in the context of adaptation planning and risk based decision making under uncertainty. The paper is very well written and structured. It will surely make an importnat contribution to the field. However, some aspects need further consideration and explanation. The most important ones are:

The rationale and details of the appraoch to determine future extreme discharges is not described comprehensively and is hard to follow at times. Reference is made to the paper Dittes et al. 2017a which is however still under review by another Journal. It would be very helpful to provide more details on the background, e.g. how the standard deviation for the hidden uncertainty is quantified, what are the underlying assuptions.

The appraoch of backwards induction optimization (page 5) should be introduced more in detail. Particulary the context that system performance is evaluated by taking data into account which is available by some poit in the future needs some additional explanation. In your study you use discarchges based on climate projections which are available today. The actually observed discharges in the future may differ from these projections, and thus may give different results and recommendations. Is this reflected by the uncertainty range of discharge projections from the climate scenarios?

The whole paragraph (page 14, ll 3-15) is not very clear. It includes a number of statements for which the basis is not comprehensible.

Your discussion of how the damage potential p 17 ll 23 - 30 should be extended by taking the broader perspective of costs of natural hazards introduced by Kreibich et al. 2014 into consideration. (Kreibich, H., van den Bergh, J. C. J. M., Bouwer, L. M., Bubeck, P., Ciavola, P., Green, C., Hallegatte, S., Logar, I., Meyer, V., Schwarze, R. and Thieken, A. H.: Costing natural hazards, Nature Clim. Change, 4(5), 303–306, doi:10.1038/nclimate2182, 2014.)

Additional minor comments and suggestions are included in the annotaded PDF.

Overall, I recommend the paper for publication in NHESS subject to minor revisions.

Please also note the supplement to this comment:
https://www.nat-hazards-earth-syst-sci-discuss.net/nhess-2017-399/nhess-2017-399-RC1-supplement.pdf

**Supplement:**

**Risk-based flood protection planning under climate change and modelling uncertainty: a pre-alpine case study**

Beatrice Dittes[1], Maria Kaiser[2], Olga Špačková[1], Wolfgang Rieger[2], Markus Disse[2], Daniel Straub[1]

[1]Engineering Risk Analysis Group, Technische Universität München, München, 80333, Germany
5  [2]Chair of Hydrology and River Basin Management, Technische Universität München, München, 80333, Germany

*Correspondence to*: Beatrice Dittes (beatrice.dittes@tum.de)

**Abstract**

Planning authorities are faced with a range of questions when planning flood protection measures: is the existing protection adequate for current and future demands or should it be extended? How will flood patterns change in the future? How should
10  the uncertainty pertaining to this influence the planning decision, e.g. for delaying planning or including a safety margin? Is it sufficient to follow a protection criterion (e.g. to protect from the 100-year flood) or should the planning be conducted in a risk-based way? How important is it for flood protection planning to accurately estimate flood frequency (changes), costs and damages? These are questions that we address for a medium-sized pre-alpine catchment in southern Germany, using a sequential Bayesian decision making framework that quantitatively addresses the full spectrum of uncertainty. We evaluate
15  different flood protection systems considered by local agencies in a test study catchment. Despite large uncertainties in damage, cost and climate, the recommendation is robustly for the most conservative approach. This demonstrates the feasibility of making robust decisions under large uncertainty. Furthermore, by comparison to a previous study, it highlights the benefits of risk-based planning over a planning of flood protection to a prescribed return period.

**1    Introduction**

20  Technical flood protection measures have long life times of, on average, 80 years (Bund / Länder-Arbeitsgemeinschaft Wasser, 2005). The uncertainty over such a long planning horizon is large, both in terms of climatic and socio-economic development. It is thus not trivial for planning authorities to take decisions on flood protection planning that are economical while not leading to excessive losses or high adjustment costs. I, it is important to consider costs – in construction, adjustment and flood damages – over the entire measure life time.

25       Ideally, the planning of flood protection infrastructure is performed through a risk-based approach. Thereby, potential damages are considered in the decision-making process. Considering that the annual maximum discharge $Q$ is the main driver for flood damages, the annual flood risk in year $t$, $r_t$, is defined as (e.g. Merz et al., 2010a)

$$r_t = \int\limits_0^\infty f_Q(q) d_t(q) \, dq \tag{1}$$

where $f_Q(q)$ is the probability density function (PDF) of the annual maximum discharge and $d_t(q)$ is the damage associated with the flood discharge $q$ in year $t$. If decisions are based on a Cost-Benefit Analysis (CBA), the optimal flood protection strategy $s$ is the one that minimizes the sum of risks and costs over the life-time of the protection system (Špačková and Straub, 2015):

$$s^{opt} = \arg\min_s \left( c^{tot}(s) + r^{tot}(s) \right), \tag{2}$$

[revised manuscript text omitted]

---

## Referee Comment (RC2) · Anonymous Referee #2 · 9 Feb 2018

Risk-based flood protection planning under climate change and modelling uncertainty: a pre-alpine case study
Author(s): Beatrice Dittes, Maria Kaiser, Olga Špačková, Wolfgang Rieger, Markus Disse, and Daniel Straub MS No.: nhess-2017-399 MS Type: Research article

This is an extremely well written paper on a very important challenge in adaptation planning – planning under deep uncertainty with flexible or inflexible strategies – also addresses adaptive management. The paper does not provide enough detail to let the reader really understand its contribution. Therefore a revision is suggest. More details below.

In the full review and interactive discussion the referees and other interested members of the scientific and technical communities are asked to take into account all of the following aspects:

1. Does the paper address relevant scientific and/or technical questions within the scope of NHESS? Yes, paper is about a natural hazard, consequences, management.

2. Does the paper present new data and/or novel concepts, ideas, tools, methods or results? Paper addresses one of the fundamental challenges of adaptation planning to climate change – how to develop strategies under deep uncertainty. They go beyond just the uncertainty of the climate change, but also the losses. Their approach builds upon applications in related fields.

3. Are these up to international standards? Yes.

4. Are the scientific methods and assumptions valid and outlined clearly? This is the drawback to this paper. The methods (Bayesian analysis and backwards induction optimization) are summarized very briefly with not enough information for the non-expert to fully understand them. While references are given, it is suggested that more detail be provided. If this is not possible due to space limitations, then more attention has to be given to them in the case study so the interested reader can see the strength of the approaches so she/he can pursue these topics in more detail.

   One challenge is that a major source of uncertainty is ignored – the emission scenario. Here they only assumed one – how can method be used if planning done more realistically under multiple emission scenarios ?

5. Are the results sufficient to support the interpretations and the conclusions? Again, more detail could have been provided. The authors determined the effectiveness of each strategy and then evaluated their performance under the uncertainties of damages and discharges. It is not clear to me why just enumeration and evaluation of all the possible

sets of strategies without the optimization model would also have been effective as small number of options. Thus would have been useful to understand the value of the optimization model. Also, the discussion of the results almost seem similar to results of conventional scenario analysis – what strategy works most reasonably over all the scenarios. Perhaps this was just a check of the results.

6. Does the author reach substantial conclusions? Generally – see #5.

7. Is the description of the data used, the methods used, the experiments and calculations made, and the results obtained sufficiently complete and accurate to allow their reproduction by fellow scientists (traceability of results)? See #5. More detail may be needed. Some figures are unclear – eg. In Figure 10, the low period discharges in many years seem higher than the high period discharged. What are the x-axis units in Table 6. Also, the term 'flexibility parameter" is used but not defined. Also, it is not clear how flexibility was considered.

8. Does the title clearly and unambiguously reflect the contents of the paper? Yes.

9. Does the abstract provide a concise, complete and unambiguous summary of the work done and the results obtained? Yes.

10. Are the title and the abstract pertinent, and easy to understand to a wide and diversified audience? Yes.

11. Are mathematical formulae, symbols, abbreviations and units correctly defined and used? If the formulae, symbols or abbreviations are numerous, are there tables or appendixes listing them? Yes.

12. Is the size, quality and readability of each figure adequate to the type and quantity of data presented? Yes.

13. Does the author give proper credit to previous and/or related work, and does he/she indicate clearly his/her own contribution? Yes. I suggest that it may be useful to compare this method to other methods of DMUUC such as Robust Decision Making, Decision Scaling, Dynamic Adaptation Pathways and Policies. This would be interesting.

14. Are the number and quality of the references appropriate? Yes. But see #13.

15. Are the references accessible by fellow scientists? Generally yes.

16. Is the overall presentation well structured, clear and easy to understand by a wide and general audience? Yes with above comments.

17. Is the length of the paper adequate, too long or too short? Adequate.

18. Is there any part of the paper (title, abstract, main text, formulae, symbols, figures and their captions, tables, list of references, appendixes) that needs to be clarified, reduced, added, combined, or eliminated? Yes, see above.

19. Is the technical language precise and understandable by fellow scientists? Yes.

20. Is the English language of good quality, fluent, simple and easy to read and understand by a wide and diversified audience? Yes. This a very well-written paper.

21. Is the amount and quality of supplementary material (if any) appropriate? It seems none was supplied.

---

## Author Comment (AC1) · 17 Mar 2018

**Author comment on the comment of anonymous referee #1**

The authors would like to thank the referee for the thoughtful and detailed comments. We respond in the following, with referee comments highlighted in *blue*.

General comment of the reviewer:

> *"The paper 'Risk based flood protection planning under climate change and modelling uncertainty: a pre-alpine case study' by B. Dittes et al. applies a framework for quantitative, probabilistic flood protection planning to a real decision making problem of flood protection strategies. This framework considers climatic uncertainties by incorporating non-stationarity and accounts for flexibility of the flood protection system in a sequential Bayesian approach. The planning problem investigated considers four alternative protection strategies with different safety levels for a city in a pre-alpine catchment. The topic is of high relevance in the context of adaptation planning and risk based decision making under uncertainty. The paper is very well written and structured. It will surely make an important contribution to the field. [...]Overall, I recommend the paper for publication in NHESS subject to minor revisions."*

The referee highlighted three *"aspects [that] need further consideration and explanation"*. These featured also in the minor comments and suggestions in the annotated PDF. The comments and corresponding changes in the manuscript are discussed in the following. In addition, we have now made the referenced papers that are still under review, (Dittes et al., 2017a) and (Dittes et al., 2017b), available on our webpage and included the link in the respective citations:

*era.bgu.tum.de/fileadmin/w00bkd/www/Papers/2017_Dittes_managing_uncertainty.pdf* and

*era.bgu.tum.de/fileadmin/w00bkd/www/Papers/2018_Dittes_et_al_Climate_uncertainty_in_flood_protection_planning.pdf*.

Minor referee suggestions that are not mentioned in the following have been changed in the manuscript directly (e.g. spelling mistakes or rephrasings/explanations of less than one sentence).

**1) Uncertainty modelling**

> *"It would be very helpful to provide more details on the background, e.g. how the standard deviation for the hidden uncertainty is quantified, what are the underlying assumptions."*

We recognize that the description of uncertainty modelling was not sufficiently clear and we have expanded on this at various places, as listed in the following along with the corresponding referee annotations.

> p4 l8: *"This differentiation of uncertainty sources should be discussed also within the context of uncertainty due to natural variability and epistemic uncertainty, e.g. in Merz, B. and Thieken, A. H.: Separating natural and epistemic uncertainty in flood frequency analysis, Journal of Hydrology, 309(1–4), 114–132, doi:10.1016/j.jhydrol.2004.11.015, 2005."*

Yes, the classification into aleatory and epistemic uncertainty is an important one. We added the following passage: *"Learning the PDF of **θ** from projections is more intricate since uncertainties from climate modelling must be accounted for. It is common to categorize uncertainty into aleatory uncertainty (natural variability), which cannot be reduced, and epistemic uncertainty, which can be reduced with more knowledge (Merz and Thieken, 2005). To account for the limited information available in typical flood protection planning problems, we here categorize the climatic uncertainties according to the following categories:[…]"*

> p4 l9 / p8 l1-6: *"In Fig. 1 this uncertainty is constant over time which means that the spread of the ensemble remains constant over time. Is this a realistic? Usually the ensemble spread increases with longer projection periods."* / *"This Fig. also shows that the ensemble spread is not constant over time, doesn't it? cf your statement on p4 l9 and Fig. 1."*

This appears to be a confusion: Fig. 1 shows internal variability vs. hidden uncertainty, it does not show ensemble spread. It is the internal variability of individual projections that is assumed to be constant (based on the data; it is not a necessary assumption). No assumptions are made on the spread of the ensemble, which does indeed increase with increasing projection horizon.

> p4 l15 / p4 l23: *"What is the basis for this assumption? Please explain more in detail. On which basis is the standard deviation quantified?"* / *"This reference is still under review. Please include the main points of the rationale also in this paper to make it more intelligable for the reader."*

We expanded and re-formulated the corresponding paragraph as follows: *"In Fig. 1, we show the hidden uncertainty and internal variability over the projection horizon for one particular projection (CCLM, see Sect. 3.3). Note that this hidden uncertainty is a rough estimate for the situation in Rosenheim based on literature (Bosshard et al., 2013; Dittes et al., 2017b; Hawkins and Sutton, 2011; Maraun, 2013). It is given as a share of total uncertainty (in terms of the variance). The hidden uncertainty is included in the analysis by considering the discharge data from projections as uncertain, with a standard deviation $\sigma_t^{(hidden)}$ that is the square root of this variance. We conservatively assume that the hidden uncertainty is fully correlated among*

*all years, since this limits the information included in the data. Mathematically this is achieved by modeling the uncertainty in year $t$ as $\times \sigma_t^{(hidden)}$, where $z$ is a standard normal random variable with PDF $\varphi$. The uncertainty is included in the likelihood $f_{Q|\boldsymbol{\theta}}(\boldsymbol{q}|\boldsymbol{\theta})$ of Eq. (3) and then integrated out:*

$$f_{Q|\boldsymbol{\theta}}(\boldsymbol{q}|\boldsymbol{\theta}) = \int_{-\infty}^{\infty} \left[ \prod_{t=1}^{Y} f_{Q|\boldsymbol{\theta}}\left(q_t - z \times \sigma_t^{(hidden)} \middle| \boldsymbol{\theta}\right) \right] \times \varphi(z) \, dz, \qquad (4)$$

*Visible uncertainties are included in different ways, e.g. the internal variability is a natural component of Eq. (4) through $q_t$, whereas the ensemble spread is inherent in combining the parameter PDFs $f_{\boldsymbol{\theta}|Q}(\boldsymbol{\theta}|\boldsymbol{q})$ from different members of a projection ensemble. For this combination, we apply the concept of effective projections (Pennell and Reichler, 2011; Sunyer et al., 2013), whereby a projection ensemble is split into multiple sets of 'effective projections'. We multiply the PDFs of the members within one set and average in between sets to obtain a joint parameter PDF. Full details of the implementation can be found in (Dittes et al., 2017b)."*

p12 l1: *"As in comment made previously. Please provide more details on this here."*

We elaborate on how the hidden uncertainty was estimated for the case study location: *"The quantitative estimate of the hidden uncertainty was taken from (Dittes et al., 2017b). It is based on the fact that Rosenheim is in a pre-alpine location, with extreme floods mostly driven by extreme summer precipitation. Using literature concerning the shares of different climatic uncertainties under extreme summer precipitation and in pre-alpine catchments (Bosshard et al., 2013; Hawkins and Sutton, 2011; Maraun, 2013), we obtained a rough estimate of the shares of various uncertainties for Rosenheim. Because the projection ensemble available for the location is based on one forcing scenario, one downscaling technique and one hydrological model only, the corresponding variance shares were together used as the 'hidden' uncertainty. The analysis in (Dittes et al., 2017b) showed that changes in the size of the hidden uncertainty have only a minor impact on the planning recommendation and a rough estimate is thus acceptable."*

p15 l27 / p15 l32: *"This is not quantified. This uncertainty is considered by using different damage models. Only on hydraulic model is used. You should make more clear how you consider these sources of uncertainty." / "You should comment how complete your assessment of uncertainty actually is."*

We elaborate: *"The recommendation for a flood protection system in Rosenheim results from a modeling sequence. The uncertainty is handed over and increased from model to model: from climate forcing uncertainty down to the hydrodynamic model and damage modeling. The climatic uncertainties as well as the internal variability are incorporated into the decision making framework by the means described in Sect. 2.1. Notably, when only one model was used at a certain step in the modeling sequence (e.g. only one forcing scenario was used), the potential for greater model spread if more models had been used is included via an estimate of the so-called 'hidden uncertainty'. No such estimate was made for the hydrodynamic model, which was simply calibrated based on recorded water levels of a flood event in 2001 and validated for the flood event of 2013 in Rosenheim (viz. Fig. 7). While the validation was*

*successful, land use changes and sedimentation of the Mangfall River may alter the discharge and water levels in the future, which we did not consider in this study. However, we are confident that such an additional uncertainty does not influence the protection decision, on the one hand because the results are generally robust with respect to changes in the hidden uncertainties (see also (Dittes et al., 2017b)), on the other hand because the recommendation already is for the most protective system."*

As requested by the referee, we added the following figure comparing the simulated extent of the 2013 flood with areal photographs of the event. The figure was added in Sect. 3.5.4, where the validation of the hydrodynamic model is first discussed.

[Figure]

**Figure 7.** Comparison of actual and simulated flood extent for the 2013 flood in Rosenheim.

**2) Decision framework**

*"The rationale and details of the approach to determine future extreme discharges is not described comprehensively and is hard to follow at times. [...] The approach of backwards induction optimization (page 5) should be introduced more in detail. Particularly, the context that system performance is evaluated by taking data into account which is available by some point in the future needs some additional explanation.*

*In your study you use discharges based on climate projections which are available today. The actually observed discharges in the future may differ from these projections, and thus may give different results and recommendations. Is this reflected by the uncertainty range of discharge projections from the climate scenarios? The whole paragraph (page 14, ll 3-15) is not very clear. It includes a number of statements for which the basis is not comprehensible."*

The main point here is that the updating with future discharge is *probabilistic*, that is, future discharges are randomly generated according to their prior probability distribution and uncertainty. The prior distribution is learned using the climate projections, but the future discharge samples resulting from them are not deterministic. It is clear from multiple annotations of the referee – mainly on page 5, where the decision framework is first introduced – that the referee thought we used the projections directly for updating, thus falsely assuming a deterministic future. Hence, we modified the description on page 5 as follows:

*"Flood protection is a dynamic process, as illustrated in Fig. 2: A flood protection system is implemented initially and later revised, based on data (e.g., discharge observations) that becomes available in the future. These future discharge observations are not yet known, hence for planning purposes they have to be simulated probabilistically, as described in the next paragraph. The damages caused by discharges in the future depend on the protection system that will then be in place. The risk is defined as the expected damages, i.e. the sum of the damages associated with each future scenario, weighted by the probability of that scenario. Ultimately, the sum of the two monetary quantities, risks and costs, is to be minimized over the measure life-time following Eq. (2). If the demand has changed based on the new observations, it may be necessary or desirable to adjust the protection capacity. The cost for both the initial implementation of the protection system and for adjustments depends on the system flexibility: a more flexible system decreases adjustment costs, but this saving must be balanced with potentially higher costs of implementing a flexible system initially. When there is large uncertainty, it becomes more likely that a design has to be adjusted later on, as more information becomes available. To take these aspects into account, we have developed a quantitative decision framework that considers planning as a sequential process. It accounts for the system flexibility and the future learning process through Bayesian updating of the initial PDF of parameters, $f_{\Theta|Q}(\theta|q)$ (Sect. 2.1.), with new information in the future (Dittes et al., 2017a). It evaluates, which flood protection system is recommendable based on the uncertainty in extreme discharge, described by $f_{\Theta|Q}(\theta|q)$, and the flexibility of the considered*

*flood protection systems. As will be shown in Sect. 3.5, the flexibility is intrinsic in the measure costs in this case study.*

*The PDF $f_{\Theta|Q}(\theta|q)$ contains the information from the currently available data: discharge projections as well as their uncertainty (through Eq. (4)). Future discharges are randomly generated from this PDF, creating a multitude of 'possible futures'. At a first revision point (e.g. 30 years into the measure life time), for each 'possible future' the PDF is updated with the discharges that were simulated to have been observed by then and a decision is made on whether the protection has to be adjusted. This process is repeated for several revision steps, leading to a decision tree with alternating adjustment decisions and observation periods (see Fig. 3). To find the optimal initial protection decision based on this tree – that is, the protection decision which minimizes the sum of life-time risks and costs – we use the technique of Backwards Induction Optimization (Raiffa and Schlaifer, 1961). The technique works by first determining the system that should be installed at the last adjustment, depending on the existing protection and observations (data) available by then. This is a deterministic problem, since at the last adjustment all the data has been collected. The evaluation is done for all possible futures and they are weighted by their probability based on the PDF. The obtained recommendation for the last adjustment is then used to find the system that should be installed at the second to last adjustment and so forth until arriving at a recommendation for the system that should be installed initially."*

**3) Damage potential estimation and individual results**

> *"Your discussion of how the damage potential p 17 ll 23 - 30 should be extended by taking the broader perspective of costs of natural hazards introduced by Kreibich et al. 2014 into consideration. (Kreibich, H., van den Bergh, J. C. J. M., Bouwer, L. M., Bubeck, P., Ciavola, P., Green, C., Hallegatte, S., Logar, I., Meyer, V., Schwarze, R. and Thieken, A. H.: Costing natural hazards, Nature Clim. Change, 4(5), 303–306, doi:10.1038/nclimate2182, 2014.)"*

We have read the reference with great interest and added to the discussion: *"The costing of natural hazards is a challenging area and the considerations given underline the need for integrated flood protection, where the cost and risk assessment cycle are linked. A comprehensive framework to do so has been proposed by Kreibich et al. (2014)."*

> p10 l13: *"Why did you not use CORINE 2012?"*

The CLC2012 data set was released at the beginning of 2017. At that time our damage calculations were already completed. However, since we get overwhelmingly agreeing protection recommendations also with differing land cover data (viz. Tab. 9), we think that using the new set would not have changed the results.

> p11 l24: *"Is this the way how the uncertainty due to loss estimation is considered in the framework?"*

Yes, the uncertainty due to loss estimation is considered by using three different damage models: RAM with ATKIS / CLC and SDAM. It is established that the recommendation is robust to the damage model (viz. Tab. 9).

> p2 l5 / p12 l7 / p12 l25: *"On which basis is the discounting done?" / "It is not clear how the discounting is implemented in the framework and on which basis it is calculated" / "What is the basis for this assumption?"*

We now clarify at the first mention of discouting (p2 l5) that it is done on an annual basis. The mathematical description of the discounting is given in Eq. (5), which we reference also at p2 l5. The chosen discouting rate of 2% corresponds to the lower bound for technical flood protection proposed in the literature (Bund / Länder-Arbeitsgemeinschaft Wasser, 2005). The dependence of protection recommendation on the choice of discount rate is studied in (Dittes et al., 2017a).

> p12 l22: *"3 m dike heightening to increase the disharge capacity from 360 to 480 m³/s? Could you give some details about the cross section geometries of the Mangfall river in Rosenheim to understand these figures better?"*

The river is about $30\ m$ wide. Thus, the section area between the dikes is $(30 + s \times h) \times h$, where $h$ is the height of the dikes and $s$ their slope (1/2 for most of the local dikes). Letting $\Delta_A$ be the difference in area and $x$ the dike heightening,

$$\Delta_A = \left(30 + \frac{h+x}{2}\right) \times (h+x) - \left(30 + \frac{h}{2}\right) \times h$$

$$= \left(30 + \frac{2 \times h + x}{2}\right) \times x$$

$$\leftrightarrow 0 = x^2 + 2 \times (30 + h) \times x - 2 \times \Delta_A$$

$$\leftrightarrow x = -(30 + h) + \sqrt{(30 + h)^2 + 2 \times \Delta_A}$$

Using a flow velocity of $1\ m/s$, $\Delta_A = 480\ m^2 - 360\ m^2 = 120\ m^2$. At a height $h$ of the existing dikes of $\sim 4\ m$, this leads to the stated result. As we only do a rough cost estimation and the recommendation is quite robust to it (viz. Tab. 9), we do not go into these details in the main text.

> p14 l3/4/10: *"How are [...] calculated" / "This whole paragraph is not very clear and hard to follow at times. A number or statements are made for which the basis is not clear. Please revise. You may also think of a better way to illustrate your results, e.g. by using a chart comparing cost-benefit relations for the different scenarios."*

We considered charts but feel that the results do not lend themselves for this – a line in a cost-benefit-chart for example would either have to cover the three damage models or the five building cost scenarios, and since there is no innate order between these one would just see a confusing squiggle. However, we do agree that the paragraph could be clearer and have therefore completely rewritten it, as well as condensed the information into one table (see following page) only: *"The expected sum of life-time costs and risks is given in Tab. 9, with the expected life-time costs individually stated in brackets. The life-time risks are calculated using Eq. (5). They are independent of measure building costs yet dependent on the system that is initially implemented. Let us first look at the damage model SDAM (which best fitted the damages of the 2013 flood, see Sect. 3.5.4) used with the reference building costs (the 'buest guess' for the building costs, see Sect. 3.6). The light blue coloring indicates that S4 is recommended for initial implementation. Thus, the expected life-time cost is the same as the initial building cost, 25.0 M €, since no adjustments are possible. The sum of life-time costs and risks is 42.6 M €. The table also shows results for the two other damage models (RAM ATKIS and RAM CLC) as well as the four other scenarios of initial building cost. When S3 is recommended for initial implementation (darker blue), the expected cost comprises the initial building cost and the expected cost of adjustment to S4 (probability of needing to adjust to S4 × cost of adjusting to S4). For SDAM, the probability of needing to adjust from S3 to S4 at a later point, if S3 was chosen initially, is 58%. For RAM using the ATKIS land cover, this probability is just 3% due to the very low damage estimates – probably a strong underestimation, as discussed in Sect. 3.5.4. When S1 is implemented initially, our computations show a residual risk of €124 M for SDAM. Thus, it is clearly better to follow the recommendation of implementing S4."*

*Table 9. Life-time costs + risks (in brackets: life-time costs only) [M €] associated with the optimal protection strategy*

| Build costs \ Damage model | SDAM | | RAM ATKIS | | RAM CLC | |
|---|---|---|---|---|---|---|
| Reference | 42.6 | (25.0) | 27.8 | (25.0) | 47.8 | (25.0) |
| Higher polder costs | 55.6 | (38.0) | 32.0 | (8.8) | 60.8 | (38.0) |
| Very high polder costs | 70.2 | (40.1) | 32.7 | (9.5) | 85.8 | (63.0) |
| Higher costs 1m initially | 46.6 | (29.0) | 31.8 | (29.0) | 51.8 | (29.0) |
| Very high costs 1m initially | 49.6 | (32.0) | 34.8 | (32.0) | 54.8 | (32.0) |

p15 l5: *"Could you indicate the values from which you calculate these 28 %? Also from Fig. 8 it is not intuitive to understand how you derive this statement."*

For the protection system S4, damages are interpolated starting from the simulated discharge of $614 \ m^3/s$. Since the local 100-year discharge estimate is $480 \ m^3/s$, this corresponds to a 28% margin. Due to the large spacing of simulated discharges and discrepancies between the damage models, this is a rough estimate. We therefore decided now to use '~30%'. We re-formulated the passage to read *"For the protection system S4, damages start occuring above the simulated discharge of 614 m³ s⁻¹ (viz. Fig. 8). Thus, recommending S4 corresponds to recommending a safety margin of ~ 30 % with respect to the 100-year flood estimate of 480 m³ s⁻¹".*

**References**

Bosshard, T., Carambia, M., Goergen, K., Kotlarski, S., Krahe, P., Zappa, M. and Schär, C.: Quantifying uncertainty sources in an ensemble of hydrological climate-impact projections, Water Resources Research, 49(3), 1523–1536, doi:10.1029/2011WR011533, 2013.

Bund / Länder-Arbeitsgemeinschaft Wasser: Leitlinien zur Durchführung dynamischer Kostenvergleichsrechnungen, 7th ed., edited by WI-00.3 DWA-Arbeitsgruppe, Berlin., 2005.

Dittes, B., Špačková, O. and Straub, D.: Managing uncertainty in design flood magnitude: Flexible protection strategies vs. safety factors, Journal of Flood Risk Management, submitted [online] Available from: https://www.era.bgu.tum.de/fileadmin/w00bkd/www/Papers/2017_Dittes_managing_uncertainty.pdf, 2017a.

Dittes, B., Špačková, O., Schoppa, L. and Straub, D.: Managing uncertainty in flood protection planning with climate projections, Hydrology and Earth System Sciences, under review, 2017b.

Hawkins, E. and Sutton, R.: The potential to narrow uncertainty in projections of regional precipitation change, Climate Dynamics, (37), 407–418, 2011.

Kreibich, H., Van Den Bergh, J. C. J. M., Bouwer, L. M., Bubeck, P., Ciavola, P., Green, C., Hallegatte, S., Logar, I., Meyer, V., Schwarze, R. and Thieken, A. H.: Costing natural hazards, Nature Climate Change, 4(5), 303–306, doi:10.1038/nclimate2182, 2014.

Maraun, D.: When will trends in European mean and heavy daily precipitation emerge?, Environmental Research Letters, 8(1), 14004, doi:10.1088/1748-9326/8/1/014004, 2013.

Pennell, C. and Reichler, T.: On the effective number of climate models, Journal of Climate, 24(9), 2358–2367, doi:10.1175/2010JCLI3814.1, 2011.

Raiffa, H. and Schlaifer, R.: Applied Statistical Decision Theory, 5th ed., The Colonial Press, Boston., 1961.

Sunyer, M. A., Madsen, H., Rosbjerg, D. and Arnbjerg-Nielsen, K.: Regional interdependency of precipitation indices across Denmark in two ensembles of high-resolution RCMs, Journal of Climate, 26(20), 7912–7928, doi:10.1175/JCLI-D-12-00707.1, 2013.

---

## Author Comment (AC2) · 17 Mar 2018

**Author comment on the comment of anonymous referee #2**

The authors would like to thank the referee for the thoughtful comments. Much of the reviewers notes were positive. We respond to suggestions for improvement in the following, with referee comments highlighted in *blue*.

1) *"The methods (Bayesian analysis and backwards induction optimization) are summarized very briefly with not enough information for the nonexpert to fully understand them. While references are given, it is suggested that more detail be provided." / "[...] it is not clear how flexibility was considered."*

We recognise that this was a weak point in the initial manuscript and have extended our description of the methods:

*"Flood protection is a dynamic process, as illustrated in Fig. 2: A flood protection system is implemented initially and later revised, based on data (e.g., discharge observations) that becomes available in the future. These future discharge observations are not yet known, hence for planning purposes they have to be simulated probabilistically, as described in the next paragraph. The damages caused by discharges in the future depend on the protection system that will then be in place. The risk is defined as the expected damages, i.e. the sum of the damages associated with each future scenario, weighted by the probability of that scenario. Ultimately, the sum of the two monetary quantities, risks and costs, is to be minimized over the measure life-time following Eq. (2). If the demand has changed based on the new observations, it may be necessary or desirable to adjust the protection capacity. The cost for both the initial implementation of the protection system and for adjustments depends on the system flexibility: a more flexible system decreases adjustment costs, but this saving must be balanced with potentially higher costs of implementing a flexible system initially. When there is large uncertainty, it becomes more likely that a design has to be adjusted later on, as more information becomes available. To take these aspects into account, we have developed a quantitative decision framework that considers planning as a sequential process. It accounts for the system flexibility and the future learning process through Bayesian updating of the initial PDF of parameters, $f_{\Theta|Q}(\theta|q)$ (Sect. 2.1.), with new information in the future* (Dittes et al., 2017). *It evaluates, which flood protection system is recommendable based on the uncertainty in extreme discharge, described by $f_{\Theta|Q}(\theta|q)$, and the flexibility of the considered*

*flood protection systems. As will be shown in Sect. 3.5, the flexibility is intrinsic in the measure costs in this case study.*

*The PDF $f_{\Theta|Q}(\theta|q)$ contains the information from the currently available data: discharge projections as well as their uncertainty (through Eq. (4)). Future discharges are randomly generated from this PDF, creating a multitude of 'possible futures'. At a first revision point (e.g. 30 years into the measure life time), for each 'possible future' the PDF is updated with the discharges that were simulated to have been observed by then and a decision is made on whether the protection has to be adjusted. This process is repeated for several revision steps, leading to a decision tree with alternating adjustment decisions and observation periods (see Fig. 3). To find the optimal initial protection decision based on this tree – that is, the protection decision which minimizes the sum of life-time risks and costs – we use the technique of Backwards Induction Optimization (Raiffa and Schlaifer, 1961). The technique works by first determining the system that should be installed at the last adjustment, depending on the existing protection and observations (data) available by then. This is a deterministic problem, since at the last adjustment all the data has been collected. The evaluation is done for all possible futures and they are weighted by their probability based on the PDF. The obtained recommendation for the last adjustment is then used to find the system that should be installed at the second to last adjustment and so forth until arriving at a recommendation for the system that should be installed initially."*

2) *"One challenge is that a major source of uncertainty is ignored – the emission scenario. Here they only assumed one – how can method be used if planning done more realistically under multiple emission scenarios ?"*

There appears to be a misunderstanding: the uncertainty on the emission scenario (which we call forcing scenario) is part of the analysis, via the 'hidden uncertainty'. We have added a sentence to clarify this: " *[…] when only one model was used at a certain step in the modeling sequence (e.g. only one forcing scenario was used), the potential for greater model spread if more models had been used is included via an estimate of the so-called 'hidden uncertainty'."*

3) *"The authors determined the effectiveness of each strategy and then evaluated their performance under the uncertainties of damages and discharges. It is not clear to me why just enumeration and evaluation of all the possible sets of strategies without the optimization model would also have been effective as small number of options. Thus would have been useful to understand the value of the optimization model. Also, the discussion of the results almost seem similar to results of conventional scenario analysis – what strategy works most reasonably over all the scenarios. Perhaps this was just a check of the results."*

We hope that we interpret the referee's point correctly as asking about the distinction between a scenario-based approach versus our optimization. As such, it points back to 1) (better description of the optimization model). The key point is that our optimization takes into account the uncertainty in discharge (including climate projections on a continuous rather than scenario-based uncertainty spectrum, future updating, measure flexibility etc., as described in

1) and (Dittes et al., 2017)) but it does not account for the uncertainty in damage model or measure building cost. This is because we focussed on irreducible uncertainties (in particular, climate) when developing our methods, whereas local building costs and damage potential are informations which can be known. Because they turned out to be not so well known after all at the case study site, we made the pragmatic decision to perform our optimization for a number of damage models and building costs. We realize that the description of the results could have been clearer, which may have contributed to the confusion of the referee. Therefore we have completely rewritten it, as well as condensed the results into one table only (see below): *"The expected sum of life-time costs and risks is given in Tab. 9, with the expected life-time costs individually stated in brackets. The life-time risks are calculated using Eq. (5). They are independent of measure building costs yet dependent on the system that is initially implemented. Let us first look at the damage model SDAM (which best fitted the damages of the 2013 flood, see Sect. 3.5.4) used with the reference building costs (the 'buest guess' for the building costs, see Sect. 3.6). The light blue coloring indicates that S4 is recommended for initial implementation. Thus, the expected life-time cost is the same as the initial building cost, 25.0 M €, since no adjustments are possible. The sum of life-time costs and risks is 42.6 M €. The table also shows results for the two other damage models (RAM ATKIS and RAM CLC) as well as the four other scenarios of initial building cost. When S3 is recommended for initial implementation (darker blue), the expected cost comprises the initial building cost and the expected cost of adjustment to S4 (probability of needing to adjust to S4 × cost of adjusting to S4). For SDAM, the probability of needing to adjust from S3 to S4 at a later point, if S3 was chosen initially, is 58%. For RAM using the ATKIS land cover, this probability is just 3% due to the very low damage estimates – probably a strong underestimation, as discussed in Sect. 3.5.4. When S1 is implemented initially, our computations show a residual risk of €124 M for SDAM. Thus, it is clearly better to follow the recommendation of implementing S4."*

*Table 9. Life-time costs + risks (in brackets: life-time costs only) [M €] associated with the optimal protection strategy*

| Build costs \ Damage model | SDAM | | RAM ATKIS | | RAM CLC | |
|---|---|---|---|---|---|---|
| Reference | 42.6 | (25.0) | 27.8 | (25.0) | 47.8 | (25.0) |
| Higher polder costs | 55.6 | (38.0) | 32.0 | (8.8) | 60.8 | (38.0) |
| Very high polder costs | 70.2 | (40.1) | 32.7 | (9.5) | 85.8 | (63.0) |
| Higher costs 1m initially | 46.6 | (29.0) | 31.8 | (29.0) | 51.8 | (29.0) |
| Very high costs 1m initially | 49.6 | (32.0) | 34.8 | (32.0) | 54.8 | (32.0) |

4) *"In Figure 10, the low period discharges in many years seem higher than the high period discharged."*

This was a mixup in the description, the sentence should read *"[…]a set of relatively low discharges (blue dots) or a set of relatively high discharges (orange dots)."* (rather than *"blue"* and *"orange"* the other way round).

5) *"What are the x-axis units in Table 6"*

Table 6 shows protection strategies. Thus one could label the x-axis with "Strategy 1, Strategy 2, …" but we feel that the existing table header *"Potential protection strategies for Rosenheim"* may be sufficiently explanatory.

6) *"[…] the term 'flexibility parameter" is used but not defined."*

Yes, while flexibility was introduced in some detail, the 'flexibility parameter' was not. We adapt the sentece as follows: *"The decision to heighten dikes and walls by 1 m would correspond to a flexibility parameter of 0.7 following (Dittes et al., 2017), where 1 corresponds to full flexibility and 0 to no flexibility."*

7) *"I suggest that it may be useful to compare this method to other methods of DMUUC such as Robust Decision Making, Decision Scaling, Dynamic Adaptation Pathways and Policies."*

We briefly answer to the methods mentioned by the referee, but would like to point to (Dittes et al., 2017) for a fuller discussion of the utilized optimization framework with respect to other DMUUC methods, which we feel does not fit into the scope of the presented paper. The consideration of system performance under a broad range of possible future developments is inherent (and quantitative) in the proposed framework, as such, it leads to robust decisions. Decision Scaling and Dynamic Adaptation Pathways and Policies also lead to robust decisions, but they do so in a discrete, (semi-)qualitative way. We take a quantiative, probabilistic approach to Engineering problems and for that reason developed our optimization framework accordingly.

**References**

Dittes, B., Špačková, O. and Straub, D.: Managing uncertainty in design flood magnitude: Flexible protection strategies vs. safety factors, Journal of Flood Risk Management, submitted [online] Available from: https://www.era.bgu.tum.de/fileadmin/w00bkd/www/Papers/2017_Dittes_managing_uncertainty.pdf, 2017.

Raiffa, H. and Schlaifer, R.: Applied Statistical Decision Theory, 5th ed., The Colonial Press, Boston., 1961.